

# Assessing seasonal climate predictability using a deep learning application: NN4CAST

Víctor Galván Fraile[1], Belén Rodríguez-Fonseca[1,2], Irene Polo[1], Marta Martín-Rey[1], and María N. Moreno-García[3]

[1]Departamento de Física de la Tierra y Astrofísica, Universidad Complutense de Madrid, Madrid, Spain
[2]Instituto de Geociencias, Consejo Superior de Investigaciones Científicas, Madrid, Spain
[3]Departamento de Informática y Automática, Universidad de Salamanca, Salamanca, Spain

**Correspondence:** Víctor Galván Fraile (vgalva01@ucm.es); Belén Rodríguez-Fonseca (brfonsec@ucm.es).

**Abstract.** Seasonal climate predictions are essential for climate services, being the changes in tropical sea surface temperature (SST) the most influential drivers. SST anomalies can affect the climate in remote regions through various atmospheric teleconnection mechanisms, and the persistence/evolution of those SST anomalies can give seasonal predictability to atmospheric signals. Dynamical models often struggle with biases and low signal-to-noise ratios, making statistical methods a valuable

alternative. Deep learning models are currently providing accurate predictions, mainly in short range weather forecast. Nevertheless, the blackbox nature of this methodology makes necessary the identification of its explainability. In this context, we present NN4CAST (Neural Network foreCAST), a versatile Python deep learning tool designed to assess seasonal predictability. Starting from the original files, NN4CAST performs all the methodological steps, enabling researchers to rapidly explore the predictability of a target variable and identify its main drivers. This flexible framework allows for the quick testing of

predictive skill from different sources of predictability, making it a valuable asset for climate services. As SST is the primary source of seasonal predictability, we illustrate the application of NN4CAST to tropical and extratropical teleconnections forced by the Pacific SSTs. We show that NN4CAST can provide skillful seasonal forecasts in regions where the atmospheric response to SST anomalies is predominantly linear, such as the tropics, as well as in remote regions where the signal is highly non-linear, like Europe. Two key examples are the prediction of SST anomalies in the tropical Atlantic region during boreal spring and pre-

cipitation anomalies over the European continent during boreal autumn. The former exemplifies a predominantly linear tropical linear ENSO-Tropical North Atlantic, whereas the latter involves a highly non-linear and non-stationary ENSO-Euro-Atlantic teleconnection. Our results demonstrates NN4CAST's potential to determine and quantify the influence of specific drivers on a target variable, offering a useful tool for improving climate predictability assessments. NN4CAST enables the attribution of predictions to specific input features, helping to identify the relative importance of different sources of predictability over time

and space. In summary, NN4CAST offers a powerful framework to better characterize and understand the complex, non-linear, and non-stationary remote climate interactions.



## 1   Introduction

Seasonal forecasting attempts to provide useful information about the climate that can be expected from 1 up to 12 months. It is crucial for different sectors such as agriculture, water resource management or disaster preparedness. Seasonal predictions are mainly justified by the existence of interactions between the atmosphere and the slow, and predictable, variations in some of the components of the climate system, such as soil moisture, snow cover, stratospheric circulation, ocean heat content or Sea Surface Temperatures (SSTs). Specifically, tropical SSTs are demonstrated to be one of the most important sources of predictability at seasonal time scales due to its characteristic persistence and evolution (Shuila and Kinter III, 2006; Kirtman and Pirani, 2009; Ineson and Scaife, 2009). Earth system models (ESMs) simulate the climate system by solving the mathematical equations that represent the interactions between its components. However, they have errors due to not only the numerical resolution of the system of equations, but also to the uncertainty in the initial and boundary conditions (Hargreaves, 2010). Within the context of seasonal time scales, different challenges, whether occurring independently or in combination, may arise. For example:

1. Seasonal predictions rely on the correct representation of not only local processes such as deep convection, but large-scale ones as global teleconnections. A misrepresentation of either of them leads to poor performance in certain remote regions, where the interaction of the different signals is non-linear and seasonally dependent (Gleckler et al., 2008; Doblas-Reyes et al., 2013).

2. Multidecadal ocean variability and the Global Warming trend alter the global circulation and, thus, the way in which atmospheric teleconnections (i.e., Rossby waves) propagate, introducing non-stationarities in the system (López-Parages et al., 2015; Weisheimer et al., 2017). While ESMs can generally extrapolate well to new regimes due to their reliance on physical laws, statistical models are more dependent on the training period and may struggle to adapt to changing relationships.

3. Oceanic patterns of variability, which may provide seasonal predictability in certain regions such as the Atlantic Niño or the North-Atlantic SSTs, are not well represented by current models (Richter and Tokinaga, 2020; Roberts et al., 2021).

4. The generation of coupled simulations at these time scales demands substantial computational resources which, in turn, constrains the potential for enhancement in spatial resolution (Doblas-Reyes et al., 2013).

As an alternative to ESMs, statistical methods directly focus on the patterns and relationships between different climate variables with multiple time lags (Wilks, 2011). Another key advantage is their high computational efficiency, as they require significant computing power only during the training phase.

Artificial Intelligence (AI) is a broad field of computer science focused on developing methods and software that allow machines to perceive and respond to environmental cues using learning and decision-making principles to achieve specific goals (Russell and Norvig, 2016). Within AI, Machine Learning (ML) develops statistical algorithms that learn from data and generalise to new inputs, handling complex, high-dimensional problems better than traditional models (Samuel, 1959). Deep





Learning (DL), a subset of ML, advances neural networks by using multiple hidden layers to automatically extract complex data
patterns. Convolutional Neural Networks (CNNs), a key DL architecture, learn spatial hierarchies through convolutional layers,
reducing the need for manual feature engineering and achieving state-of-the-art performance in tasks like image recognition,
medical analysis (LeCun et al., 2015; Rawat and Wang, 2017), and, as in our case, climate predictions, where it has been proven
to be computationally efficient, as well as being a way to produce accurate weather and climate predictions (Rew et al., 2006).

    The potential of DL in this field is evident in the development of different models that are built from empirical data, rather
than through explicit theoretical equations such as dynamic models. These DL models are used by meteorological agencies and
researchers, as well as private companies, with examples such as PanguWeather, which is a 3D Earth-specific transformer mod-
ule created by the Huawei Cloud Group (Bi et al., 2022); GraphCast and NeuralGCM, developed by researchers at DeepMind
and Google (Lam et al., 2022; Kochkov et al., 2024). These are three examples of models based on meteorological data, al-
though there is a wide range of models with different architectures. These DL models produce predictions in an autoregressive
manner, which means that they generate sub-daily predictions of the state of the Earth system taking into account its temporal
evolution.

    The use of DL models to assess seasonal forecast is not so common and other ML techniques such as linear regression,
principal component analysis, correlation, and maximum covariance analysis have been widely applied with satisfactory results
(Wilks, 2014; Suárez-Moreno and Rodríguez-Fonseca, 2015; Rieger et al., 2021). Deep learning (DL) models can overcome
current limitations in seasonal prediction by improving accuracy and enabling analysis of underlying attribution mechanisms.
Some studies have applied DL to specific phenomena such as El Niño–Southern Oscillation and the Atlantic Niño (Ham et al.,
2019; Shin et al., 2022; Bachèlery et al., 2025). However, those approaches produce tailored models for each case, without
built-in explainability or generalizability to other teleconnections. In contrast, we introduce a DL framework that applies to
any climate phenomenon and explicitly analyses feature attributions to identify predictability drivers.
For these reasons, we have developed the Neural Network foreCAST (NN4CAST) application, a Python library designed
to facilitate the creation of simple deep learning models for reproducing climate teleconnections driven by different sources
of seasonal predictability. NN4CAST provides a flexible framework for non-linear statistical analysis, allowing researchers
to efficiently quantify the predictive skill of various sources of predictability. Our approach, in contrast to other machine
learning-based models, facilitates the development of a user-friendly model characterized by a simple architecture and low
computational cost. The idea behind NN4CAST is to mitigate the risk of treating deep learning methods as "black boxes"
,thereby enabling users to identify sources of predictability and assess the sensitivity of predictions to variations in the training
period and/or to the predictor region. This is an important added value of NN4CAST model compared to other DL seasonal
forecasting models (Pan et al., 2022; Watt-Meyer et al., 2024). It enables the analysis of predictability, as well as the exam-
ination of teleconnections, their modulations, the identification of windows of opportunity and the production of attributions
of the predictions over certain target regions. NN4CAST is implemented as a Python library intended for use in applications
with small or large datasets for seasonal or decadal predictions. This tool could be coupled to more complex frameworks, such
as ESMValTool, a community tool of diagnostic and performance metrics for evaluating Earth system models (Righi et al.,
2020). Scientific tools are commonly written using low-level compiled languages such as C or C++, due to their greater com-



putational efficiency compared to high-level interpreted languages such as Python. However, Python has become the dominant
programming language for ML tools, including the libraries used in NN4CAST. This highlights the importance of NN4CAST
not only for research purposes but also as a complementary tool for operational seasonal forecasting applications.

This paper introduces the NN4CAST package by first outlining the theoretical foundations of neural networks and detailing
each of its core features. It then demonstrates its utility through two concrete case studies: the prediction of Tropical North
Atlantic (TNA) SST anomalies during boreal spring (March–April–May, MAM) (Alexander et al., 2002), and the forecasting
of European precipitation anomalies in boreal autumn (October–November–December, OND) (López-Parages and Rodríguez-
Fonseca, 2012), both using Pacific SST anomalies as predictors. These two teleconnections are broadly known in the literature,
yet the potential of SST predictors to enhance seasonal forecasts of TNA SST and European rainfall has not been fully assessed.
Moreover, NN4CAST framework extends beyond ocean predictors, enabling the investigation of predictability in additional
climate variables such as soil moisture or sea ice cover. The full codebase is publicly accessible on GitHub and archived
on Zenodo (Galván Fraile et al., 2025). The remainder of the paper is organized as follows: Section 2 reviews fundamental
concepts in deep learning; Section 3 describes the NN4CAST methodology and its implementation; Section 4 presents the
two applications, highlighting both strengths and limitations; and Section 5 summarizes the main conclusions and outlines
directions for future work.

## 2   Theoretical framework

In a seasonal prediction system, a predictor field X is initialized at time $t_0$ to forecast a predictand field Y at $t_0 + \tau$, thereby
leveraging the inherent inertia of climate anomalies in X. Although numerous statistical methods exist for seasonal prediction,
the present work adopts a deep-learning framework. Construction of the neural network entails the selection of hyperparam-
eters, such as network depth, layer widths, activation functions, optimization algorithm and regularization strategies (Géron,
2022). Training is performed by minimizing the Mean Squared Error (MSE) loss, defined as:

$$MSE = \frac{1}{m} \sum_{i=1}^{m} \left( y^{(i)} - \hat{\mathbf{y}}^{(i)} \right)^2 \tag{1}$$

where $m$ is the number of samples in the dataset, $\hat{\mathbf{y}}^{(i)}$ is the predicted value of the predictand variable for the $i^{th}$ instance in
the dataset and $y^{(i)}$ is its corresponding real value (i.e., the ground truth) (Wilks, 2011). However, it is possible to modify this
objective function to any other differentiable loss function (Cuomo et al., 2022). After training by minimizing the loss function,
the model parameters are fixed and used to generate forecasts on the independent test set. Forecast skill is then quantified using
the Root Mean Squared Error (RMSE), defined as the square root of the MSE and the Anomaly Correlation Coefficient (ACC),
which is given by:

$$ACC = \frac{\sum_{i=1}^{m} y^{(i)} \cdot \hat{\mathbf{y}}^{(i)}}{\sqrt{\sum_{i=1}^{m} \left( y^{(i)} \cdot \hat{\mathbf{y}}^{(i)} \right)^2}} \tag{2}$$



where $\hat{\mathbf{y}}^{(i)}$ is the predicted anomaly of the predictand variable for the $i^{th}$ instance in the dataset, and $y^{(i)}$ is its corresponding observed anomaly, both computed relative to the same climatology (Wilks, 2008).

To achieve a more robust and unbiased evaluation of model performance, NN4CAST implements a cross-validation approach wherein the dataset is systematically partitioned into training and testing subsets across $k$ folds. This iterative process ensures that each sample is utilized for both training and validation, thereby maximizing the use of available data and providing a comprehensive assessment of the model's generalization capabilities. Furthermore, NN4CAST offers the option to perform leave-one-out cross-validation, where each individual sample is used once as a test set while the remaining samples form the

training set. This exhaustive method yields a detailed skill assessment across the entire dataset, facilitating the construction of a full-period hindcast and enhancing the reliability of predictive evaluations (Michaelsen, 1987).

To facilitate a comprehensive analysis of the mechanisms driving the predictions, eXplainable AI (XAI) techniques are employed to assess the relative importance of predictor field features for a given region of the predictand field. XAI aims to enhance the added value of AI methods by enabling the identification of the most influential predictor areas in the prediction

process. There are two main categories of methods: sensitivity methods, which assess the sensitivity of the output value to a specific predictor, and attribution methods, which determine the relative contribution of each predictor to the predictand (Guidotti et al., 2018). In this work one of the most common attribution method, known as Integrated Gradients (Sundararajan et al., 2017), is employed. This method addresses the issue of non-linear problems, where the derivative of the output with respect to the inputs is not constant. It considers a reference (baseline) vector $\hat{x}$ for which the model output is zero: $\hat{\mathbf{F}}(\hat{x}) = 0$.

The importance is computed as the product of the distance between the input within the reference point and the average of the gradients at points along the straight-line path from the reference point to the input feature. Specifically, the mathematical expression is given by:

$$R_{i,n} = (x_{i,n} - \hat{\mathbf{x}}_{\mathbf{i}}) \frac{1}{m} \sum_{j=1}^{m} \frac{\partial \hat{\mathbf{F}}}{\partial X_i} \bigg|_{X_i = \hat{\mathbf{x}}_{\mathbf{i}} + \frac{j}{m}(x_{i,n} - \hat{\mathbf{x}}_{\mathbf{i}})} \tag{3}$$

where $X_i$ are the input features, $R_{i,n}$ the relevance of feature at grid point $(i)$ for the model prediction of sample $(n)$, $\hat{\mathbf{F}}$ the

function learned by the model and $(m)$ the number of steps in the Riemann approximation (Mamalakis et al., 2022). The next section describes how this theoretical framework is implemented in the NN4CAST tool.

## 3  NN4CAST methodology & implementation

Building on the key elements of the DL methodology outlined above, the NN4CAST library integrates these approaches into a unified framework. The whole procedural workflow is depicted in Fig. 1:

· Preprocessing of the datasets according to user targets, including: data loading, selection of the region and season of interest, anomaly computation, and trend removal (I in Fig. 1).

· Construction of a deep neural network for seasonal prediction, with training performed through the minimization of the MSE (II in Fig. 1).



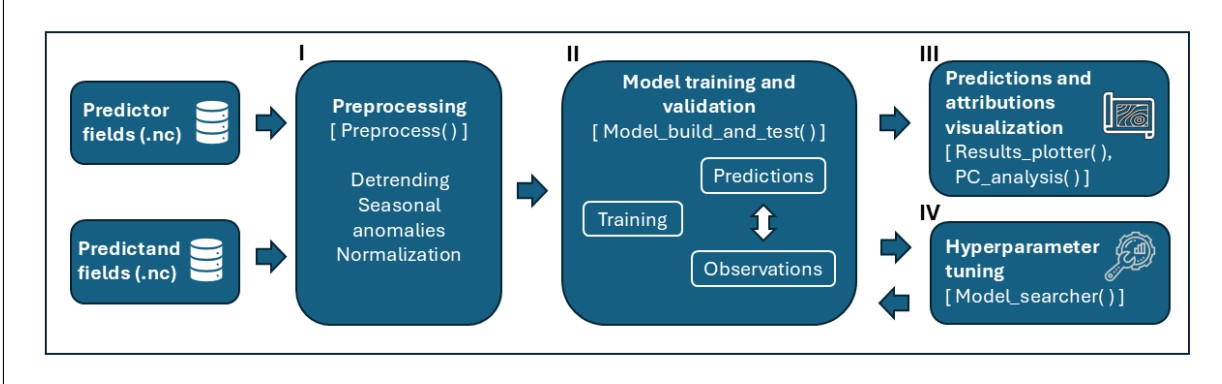

**Figure 1.** Flowchart illustrating the methodology and application workflow of the NN4CAST library, designed for processing monthly data and assessing seasonal climate predictability. The Python function names corresponding to each step are shown in brackets.

· Application of regularization techniques to mitigate overfitting (II in Fig. 1).

· Model performance evaluation using cross-validation strategies, employing different skill metrics (RMSE and ACC).

· Optimization of model hyperparameters to improve generalization and predictive performance (IV in Fig. 1).

· Attribution analysis within the XAI framework to identify the contribution of different regions of the predictor field to the model predictions (III in Fig. 1).

· Empirical Orthogonal Function (EOF) analysis of model outputs and observational data to compare dominant modes of

variability and assess the physical consistency of predictions (III in Fig. 1).

Each of the above mentioned steps is computed by the application of different Python functions designed within NN4CAST (Fig. 1. The application begins by loading predictor and predictand datasets and defining model hyperparameters. A standardized preprocessing pipeline is then applied via the `Preprocess()` function. Next, the model is built, trained, and tested using `Model_build_and_test()`, which integrates attribution routines to compute feature-importance maps for

each forecast . Once training is complete, `Results_plotter()` visualizes both the individual predictions and their associated attribution maps, making the physical drivers of skill explicit. Optionally, `Model_searcher()` can be used to optimize performance through hyperparameter tuning, and `PC_analysis()` enables EOF analysis of model outputs versus observations to assess dominant spatial modes. Finally, the entire workflow can be configured via the main module `nn4cast.predefined_classes`

The principal distinguishing feature of NN4CAST lies in its design, which specifically targets the assessment of teleconnection predictability, thereby enabling experiments aimed at addressing common challenges in seasonal climate forecasting. These problems, which were mentioned in the introduction, are solved by NN4CAST as follows:





1. The problem of representing teleconnection drivers at interannual timescales. The models created by NN4CAST can be run for different regions of the predictor field, in order to be able to identify the relative contribution of each of them.

2. The non-stationarity problem. The models created by NN4CAST can be run for different training periods in order to produce seasonal predictions in a particular periods. This allows to analyse the non-stationarity behaviour of the relationships.

3. The computational resources problem. The models created by NN4CAST are capable of performing simulations within a few minutes on a standard computer.

4. The black box nature of DL. The models created by NN4CAST have the advantage of providing insightful attributions of the predictions which allow to identify the drivers and the underlying physical mechanisms.

### 3.1 Configuration of hyperparameters and Preprocess

NN4CAST employs datasets stored in the netCDF4 format (Rew et al., 2006), a widely convention standard within the realm of Earth data science. These datasets are structured as space-time matrices and require three coordinates: time, latitude, and longitude. As NN4CAST, in its current form, is designed for seasonal time scales, the time coordinate must be defined at a monthly resolution. Additionally, NN4CAST employs a dictionary to define the hyperparameters required by the application, which are described in Table 1, giving a brief description of their functionality. Listing 1 demonstrates how to create this dictionary and save it as a .yaml file, for the case of the prediction of SST anomalies in the tropical North Atlantic from the Pacific SST anomalies. YAML is a text-based file format suitable for storing structured data, such as model configurations. Saving the dictionary as a .yaml file enables the documentation of the experiment details for easy retrieval and sharing.

| Hyperparameter name | Description |
|---|---|
| $path_{-}x, path_{-}y$ | Path to the X (predictor) and Y (predictand) datasets files. |
| $time\_limits$ | Define the time limits (in years) to select from the original datasets. They should be in the format ['yyyy', 'yyyy'], where the first year is the start and the second year is the end of the period, separated by commas. For example, ['1975','2015']. |
| $jump\_year$ | Specify whether the predictor (X) and predictand (Y) fields are drawn from the same calendar year or from different years. A value of 0 (default) indicates same-year data; any positive integer denotes the lead time in years between X and Y. For example, if X corresponds to September of year N and Y to February of year N+1, setting $jump_{-}year = 1$ relates the predictor and predictand across consecutive years. |
| $months_{-}x, months_{-}y$ | Select the months of X and Y, knowing that January=1 and December=12. |
| $name_{-}x, name_{-}y$ | Specify the variable names for the predictor and predictand fields, as defined in the original netcdf files. |
| $units_{-}x, units_{-}y$ | Set the units of the input and output fields to fill the labels in the plot. |
| $region\_predictor$ | Set the name of the predictor region to be included as information for the figures. |
| $months\_skip_{-}x, months\_skip_{-}y$ | Define the months to skip. This is necessary if several months are considered to analyse seasons involving changes of the year. For example, if X samples are from Dec-Jan-Feb and the data spans from 1950 to 2019, it is necessary to set $months_{-}skip_{-}x$ = ['1950-01', '1950-02', '2019-12'], to delete the Jan-Feb from the first year and the Dec from the last one, in order to create seasonal means that makes climatic sense. |





| | |
|---|---|
| $train\_years, validation\_years,$ $test\_years$ | Define the periods (initial year, final year) for training, validating and testing the model based on X years, using the same format as for $time\_limits$. |
| $mean\_seasonal\_method\_x,$ $mean\_seasonal\_method\_y$ | Define if computing the seasonal means (=True) or aggregates (=False). |
| $lat\_lims\_x, lon\_lims\_x,$ $lat\_lims\_y, lon\_lims\_y$ | Define the latitude-longitude regions for the predictor (X) and predictand (Y) fields. For the latitudes, it can be selected in any order (smaller first or last), whereas, for the longitudes, it is needed to put the smaller first, and either in the format: -180-(+180) or 0-360. |
| $scale\_x, scale\_y$ | Define if scaling the data, if not, set as 1 (default=1). For example, if the pressure data is given in Pa and you want to have the data in hPa, then, a scale of 100 need to be applied. |
| $regrid\_degree\_x,$ $regrid\_degree\_y$ | Define the interpolation degree for regridding the data. Values greater than 0 apply regridding, while 0 means no regridding (default=0). |
| $detrend\_x, detrend\_y$ | Define if performing a backward moving average detrend of the data. If True, it is needed to define the window size to compute the detrend iteratively. |
| $detrend\_x\_window,$ $detrend\_y\_window$ | Define the size of the past window to compute the averages to do the detrend (default=10). |
| $layer\_sizes$ | Define the number of neurons per layer of the model as a list. For example, [1024, 256, 64]. |
| $activations$ | Define the activation function of each layer of the model as a list. For example, [tf.keras.activations.elu, tf.keras.activations.elu, tf.keras.activations.elu]. |
| $kernel\_regularizer$ | Define whether using or not a regularizer. The options are: $l\_1, l\_2, l1\_l2$ or $None$ (default=['l1_l2']). |
| $learning\_rate$ | Set the learning rate of model hyperparameters update (default=0.0001). |
| $epochs$ | Set the number of epochs to train the model, knowing that there is an early stopping when no increase is reached in the validation loss on the last 25 epochs (default=2500). |
| $num\_conv\_layers$ | Set the number of convolution layers to apply (default=0). If at least 1 convolution layer is applied, you can define also the $number\_filters$, $kernel\_size$ and $pooling\_size$. |
| $use\_batch\_norm,$ $use\_initializer$ | Set to True to perform batch normalization and Kaiming He initialization (He et al., 2015) of the parameters in each layer of the model. (default=False). |
| $use\_dropout, dropout\_rates$ | Set to True to perform dropout, giving also the dropout rates as a list. (default=False). |
| $use\_init\_skip\_connections,$ $use\_inter\_skip\_connections$ | Set to True to use initial or intermediate skip connections as in the U-Net architecture (Ronneberger et al., 2015) (default=False). |
| $p\_value$ | Define the $p\_value$ used to establish the confidence interval through the T-test method for assessing statistically significant values of the model skill. |
| $outputs\_path$ | Select the directory where the plots and datasets will be saved. |

**Table 1.** Table presenting the names and detailed descriptions of the hyperparameters used in the NN4CAST application, which govern the model configuration and training process.

The next step is to preprocess the data of the predictor and predictand, in order to compute the operations given in the hyperparameters dictionary: regriding the dataset, seasonal mean anomalies, detrending, etc. It is important to note how the detrending works. It is made by applying a backward moving average (BMA) algorithm, which computes the running mean of the previous years using a sliding window and substracting it to the following value (Raffalovich, 1994; Alvarez-Ramirez et al., 2005). By this way it avoids to introduce future information in the preprocessing phase. For example, if the size of the sliding window is 20 years, the detrend will be applied from year 21 onwards. It is important to note that this detrending methodology

190



is not a standard linear detrending technique. Rather, low frequencies may also be filtered out in the process, depending on the size of the sliding windows (larger windows will filter out only the lowest frequencies). The preprocess is done by applying the function `Preprocess()`, as shown in Listing 2, returning a dictionary with the datasets needed to train, validate and test the model.

## 3.2 Model cross-validation & Explainable AI

To obtain robust and objective estimates of model performance, NN4CAST includes a cross-validation routine via the `Model_build_and_test()` function. While a standard train–validation–test split (e.g., 70%–10%–20%) is possible, cross-validation is generally recommended, as it maximizes the use of available data and reduces sensitivity to the arbitrary selection of training periods. By setting the `n_cv_folds` parameter, the dataset is partitioned into $k$ folds: in each iteration, the model is trained on $k-1$ folds and tested on the remaining one, cycling through all folds. To guard against overfitting, a random 10% of each training fold is reserved for early stopping. When $k$ equals the number of samples, this defaults to leave-one-out cross-validation, producing a complete hindcast over the full period (see Listing 3).

A core objective of NN4CAST is to facilitate the generation of insightful attributions between predictions. To compute these attributions for a specific target region of the predictand field, first set the instance to `True` to enable the computation. Then, define the region as a list of latitude and longitude ranges (setting region_importances=[[latitude_min, latitude_max], [longitude_min, longitude_max]]), as shown in Listing 3. By this way, the model will apply the Integrated Gradients methodology included in the Alibi python library (Klaise et al., 2021). This methodology benefits from the cross-validation approach in order to obtain the attributions over the entire hindcast. In the following section, the performance of NN4CAST is evaluated on two case studies that are particularly relevant for the seasonal prediction community. These applications use the default hyperparameter configuration to illustrate the practical utility and interpretability of the application.

## 3.3 Model hyperaparameters tunning

NN4CAST also offers the possibility of optimising the hyperparameters of the model in order to achieve an overall better performance. First, a dictionary of the possible values of the hyperparameters to search has to be defined, as shown in Listing 4. Then, an instance of the `Model_searcher()` functionality is created, defining also the maximum number of trials to search for the optimum hyperparameters, and applying the random search methodology (Bergstra and Bengio, 2012). The model is then evaluated following a cross-validation scheme given by the number of folds into which the dataset is divided.

## 4 Applications of NN4CAST

### 4.1 Modelling Paficic-North Tropical Atlantic SST Connection

In this section, we present a case study to illustrate the applicability of the NN4CAST framework for modelling a well-documented atmospheric teleconnection. Specifically, we focus on the relationship between sea surface temperature (SST)





```python
from tensorflow.keras import activations
import numpy as np
from nn4cast.predefined_classes import Dictionary_saver

hyperparameters =
    'path_x' = "/path/to/your/data/HadISST1_sst_1870-2019.nc",
    'path_y' = "/path/to/your/data/HadISST1_sst_1870-2019.nc",

    'time_limits' = [1900,2019],
    'jump_year' = 0,

    'lat_lims_x' = [30, -30],
    'lon_lims_x' = [120, 290],
    'lat_lims_y' = [40, -10],
    'lon_lims_y' = [-80, +20],

    'name_x' = 'sst',
    'name_y' = 'sst',

    'months_x' = [1, 2, 3],
    'months_skip_x' = ['None'],
    'months_y' = [4, 5, 6],
    'months_skip_y' = ['None'],

    'mean_seasonal_method_x' = True,
    'mean_seasonal_method_y' = True,

    'regrid_degree_x' = 2,
    'regrid_degree_y' = 2,

    'scale_x' = 1,
    'scale_y' = 1,

    'detrend_x' = True,
    'detrend_y' = True,
    'detrend_x_window' = 30,
    'detrend_y_window' = 30,

    # Neural network hyperparameters (default ones).
    'layer_sizes' = [1024,256,64,256,1024],
    'activations' = [activations.elu,activations.elu,activations.elu,
      activations.elu, activations.elu]
    'dropout_rates' = [0.1],
    'kernel_regularizer' = 'l2',
    'learning_rate' = 0.0001,
    'epochs' = 2500,
    'num_conv_layers' = 0,
    'use_batch_norm' = True,
```





```
        'use_initializer' = True,
        'use_dropout' = True,
        'use_init_skip_connections' = False,
        'use_inter_skip_connections' = False,

57
        'units_x' = ['°C'],
        'units_y' = ['°C'],
        'region_predictor' = 'Tropical Pacific',
        'p_value' = 0.1,
62      'outputs_path' = "/path/to/the/directory/Outputs_ND_sst_SO/"

        Dictionary_saver(hyperparameters)
```

**Listing 1.** Python code example demonstrating how to create an instance of the parameter and hyperparameter dictionary, and how to save it using the `Dictionary-saver` function.

```
        from nn4cast.predefined_classes import Preprocess

        dictionary_preprocess = Preprocess(dictionary_hyperparams=hyperparameters)
```

**Listing 2.** Python code example demonstrating the use of the `Preprocess` function to manipulate predictor and predictand data, and to store the processed outputs in a dictionary format.

```
        from nn4cast.predefined_classes import Model_build_and_test, Results_plotter

        outputs_cross_validation = Model_build_and_test(dictionary_hyperparams =
          hyperparameters, dictionary_preprocess=dictionary_preprocess,
5         cross_validation=True, n_cv_folds=120, plot_differences=False, importances=True,
          region_importances=[[5,25],[-55,-15]])

        Results_plotter(hyperparameters, dictionary_preprocess, rang_x=1.5, rang_y=1,
          predictions=outputs_cross_validation['predictions'],
10        observations=outputs_cross_validation['observations'],
          years_to_plot=[2015], plot_with_contours=True,
          importances=outputs_cross_validation['importances'],
          region_importances=outputs_cross_validation['region_attributed'])
```

**Listing 3.** Python code example demonstrating the use of the `Model-build-and-test` function to construct the model using selected hyperparameters, followed by training, validation, and testing through a user-defined cross-validation approach. The listing also includes the use of the `Results-plotter` function to visualize the model outputs and their attributions for the target region over the predictor field..



```
      from nn4cast.predefined_classes import Model_searcher

      params_selection =
          'pos_number_layers' = 5, # set the maximum value of fully connected layers (int).
 'pos_layer_sizes' = [16, 64, 256],# set the possible layer sizes (list).
          'pos_activations' = ["elu", "linear"],# set the possible activation functions
             (possibilities are all the ones availabe: tf.keras.layers.activations()) (list).
          'pos_dropout' = [0.0, 0.01],# set the possible dropout probabilities (list).
          'pos_kernel_regularizer' = ["l1_l2"],# set the possible kernel regularizer
(possibilities are: l1_l2, l1, l2, None) (list).
          'search_skip_connections' = False,# set if searching for skip connections, either
             intermediate or end_to_end connections (bool).
          'pos_conv_layers' = 0,# set the maximum number of convolutional layers, the
             predictor field (X) must be 2D (int).
'pos_learning_rate' = [1e-4,1e-3],# set the possible learning rates (list).

      outputs_bm_cross_validation = Model_searcher(dictionary_hyperparams=hyperparameters,
         dictionary_preprocess=dictionary_preprocess,
         dictionary_possibilities=params_selection, max_trials=10, n_cv_folds=8)
```

**Listing 4.** Python code example demonstrating the use of the `Model-searcher` function to identify the optimal set of hyperparameters within a predefined search space. The hyperparameter ranges are based on commonly reported values in the specialized literature and explored using the random search method (Bergstra and Bengio, 2012). The best combination obtained is then evaluated following the selected cross-validation scheme.

anomalies in the tropical Pacific during boreal winter (December-January–February; DJF) and SST anomalies in the tropical North Atlantic (TNA) during the subsequent spring (March-April–May; MAM). This teleconnection has been extensively studied in the literature (Enfield and Mayer, 1997; Alexander et al., 2002; Lee et al., 2008; García-Serrano et al., 2017), making

it an ideal test case to demonstrate the capabilities of NN4CAST. It is well established that El Niño events during boreal winter are typically associated with warming in the TNA region through a process commonly referred to as the atmospheric bridge between the Pacific and Atlantic Oceans (Alexander et al., 2002). Several mechanisms have been proposed to explain this teleconnection. One hypothesis involves the weakening of the subtropical high over the North Atlantic via atmospheric Rossby waves originating in the tropical Pacific and propagating through the Pacific–North America (PNA) sector (Enfield and Mayer,

1997). Another mechanism suggests that changes in the Pacific zonal circulation may influence convection in the Atlantic, thereby modulating the Atlantic meridional circulation (Klein et al., 1999). Both mechanisms ultimately lead to a weakening of the subtropical high, resulting in reduced trade winds and enhanced SST warming in the TNA. Additional theories in the literature point to upper-tropospheric responses to Pacific SST anomalies, which can trigger eastward-propagating Kelvin waves (Chang et al., 2001), as well as to a remote Gill-type response to the ENSO-related changes in the Pacific Walker

circulation (García-Serrano et al., 2017). The teleconnection between ENSO and TNA SST anomalies depends on the type and persistence of ENSO events, which determine the strength and duration of the atmospheric bridge. Additionally, the North




| Parameter | Value | Parameter | Value |
|---|---|---|---|
| **Input/Output, Temporal and Spatial Settings** | | | |
| region_predictor | Tropical Pacific | time_lims | 1900–2019 |
| lat_lims_x | 30, -30 | lon_lims_x | 120, 290 |
| lat_lims_y | 40, -10 | lon_lims_y | -80, 20 |
| **Preprocessing** | | | |
| detrend_x/y | True/True | detrend_x/y_window | 30/30 |
| mean_seasonal_method_x/y | True/True | regrid_degree_x/y | 2/2 |
| months_x | 12, 1, 2 | months_y | 3, 4, 5 |
| **Model Architecture** | | | |
| layer_sizes | 1024, 256, 64, 256, 1024 | activations | ELU |
| num_conv_layers | 0 | dropout_rates | 0.1 |
| use_dropout | True | use_batch_norm | True |
| use_initializer | True | kernel_regularizer | l2 |
| use_init_skip_connections | False | use_inter_skip_connections | False |
| **Training Settings** | | | |
| learning_rate | 0.0001 | epochs | 2500 |
| p_value | 0.1 | jump_year | 0 |
| cross_validation | True | CV strategy | Leave-One-Out |

**Table 2.** Table summarizing the selected model hyperparameters and preprocessing settings used in the simulation of DJF Pacific – MAM Tropical North Atlantic SSTs.

Atlantic Oscillation (NAO) can modulate the TNA response, as its negative phase—associated with a weaker subtropical high—can enhance the influence of El Niño on TNA SSTs (Lee et al., 2008; Czaja et al., 2002; Wu et al., 2020).

Using the NN4CAST framework, a tailored model can be efficiently developed to simulate the Pacific–Atlantic teleconnection, assess its predictive skill across various events, and explore attribution patterns for selected case studies. These patterns can then be compared against the mechanisms proposed in existing scientific studies. To analyse this teleconnection using NN4CAST, we utilise the HadISST dataset as the source of sea surface temperature (SST) data for both predictor and predictand fields (Rayner et al., 2003). The predictor region corresponds to the tropical Pacific basin [30°S–30°N; 120°E–70°W]. This region captures the core variability associated with ENSO phenomena. The predictand region represents the tropical North Atlantic [10°S–40°N; 80°W–20°E], encompassing the key area where the SST anomalies associated with this teleconnection typically manifest during the boreal spring. The parameters and hyperparameters used are the ones shown in Table 2. The same steps as defined in Listings 1, 2 and 3 are done. Specifically, first, the dictionary with the details of the simulation is created and saved in the outputs directory. Then, the preprocessing of the datasets is done by applying as optional main arguments: regriding the data to reduce the computational cost of the simulation and detrending to remove the signal associated with the





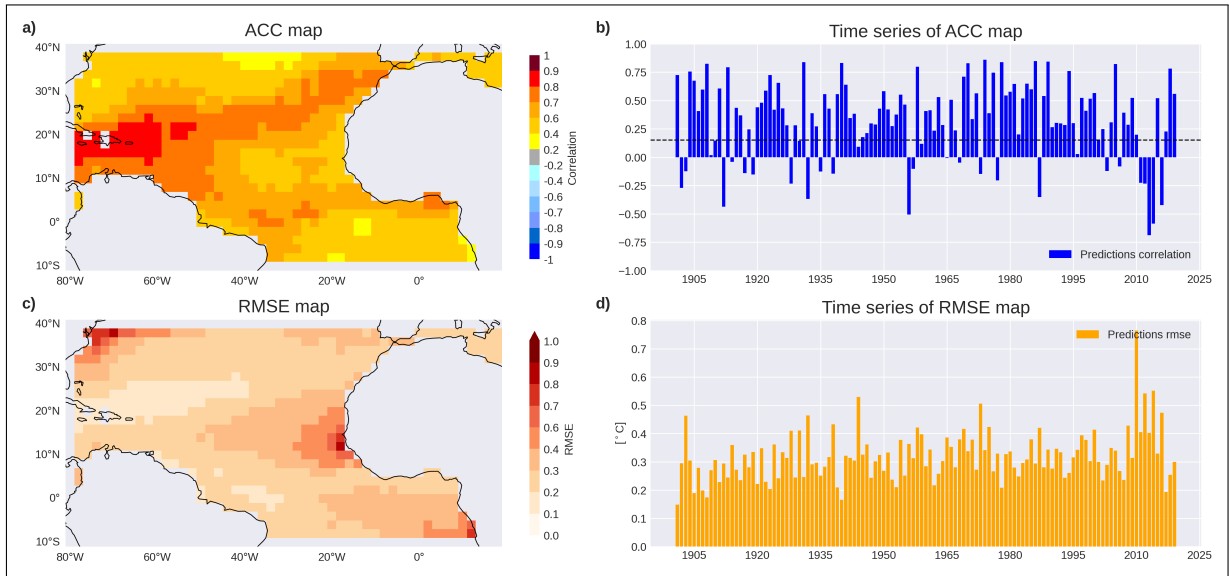

**Figure 2.** Predictability of tropical North Atlantic SST variability from tropical Pacific anomalies. Panel showing model performance metrics over the full period (1901–2019) using a leave-one-out cross-validation approach for predicting the SST anomaly field of the tropical North Atlantic during MAM, with Tropical Pacific SST from DJF as the predictor. The predictions are compared against observed MAM SST anomalies. Specifically: (a) ACC spatial map, correlating for each grid point the observed and predicted time series; (b) Time series of ACC maps, correlating for each year the observed and predicted spatial patterns; (c) RMSE spatial map; and (d) Time series of RMSE maps, computed analogously to the ACC metrics, all calculated between predicted and observed fields. The ACC (RMSE) time series show the correlation (error) between predicted and observed global mean SST anomalies over time. Statistically significant results, based on a one-tailed t-test with a significance level defined in the hyperparameters (95%), are indicated by the non-dashed regions in panel (a) and values above the dashed line in panel (b).

anthropogenic warming trend. Subsequently, the model is initialised using the hyperparameters specified in Listing 1, adopting an encoder–decoder architecture (Goodfellow et al., 2016). This design first compresses the high-dimensional predictor field into a low-dimensional latent representation, then reconstructs the target field from that embedding, facilitating efficient extraction and interpretation of the most relevant predictive features.

To obtain a robust measure of model performance, we apply a cross-validation approach. This is achieved by enabling the cross-validation argument and specifying the number of folds into which the dataset will be partitioned, as demonstrated in Listing 3. Specifically, a leave-one-out cross-validation scheme with 120 folds is employed. Furthermore, to identify the regions from which the model extracts information to generate the TNA signal, attributions for two regions are computed using the Integrated Gradients method, as outlined in Section 2: one representing the western TNA (WTNA) region [10°N–20°N; 50°W-70°W], and the other representing the southern Mauritanian-Senegalese Coastal Upwelling (SMSCU) system [10°N–20°N; 25°W-15°W].



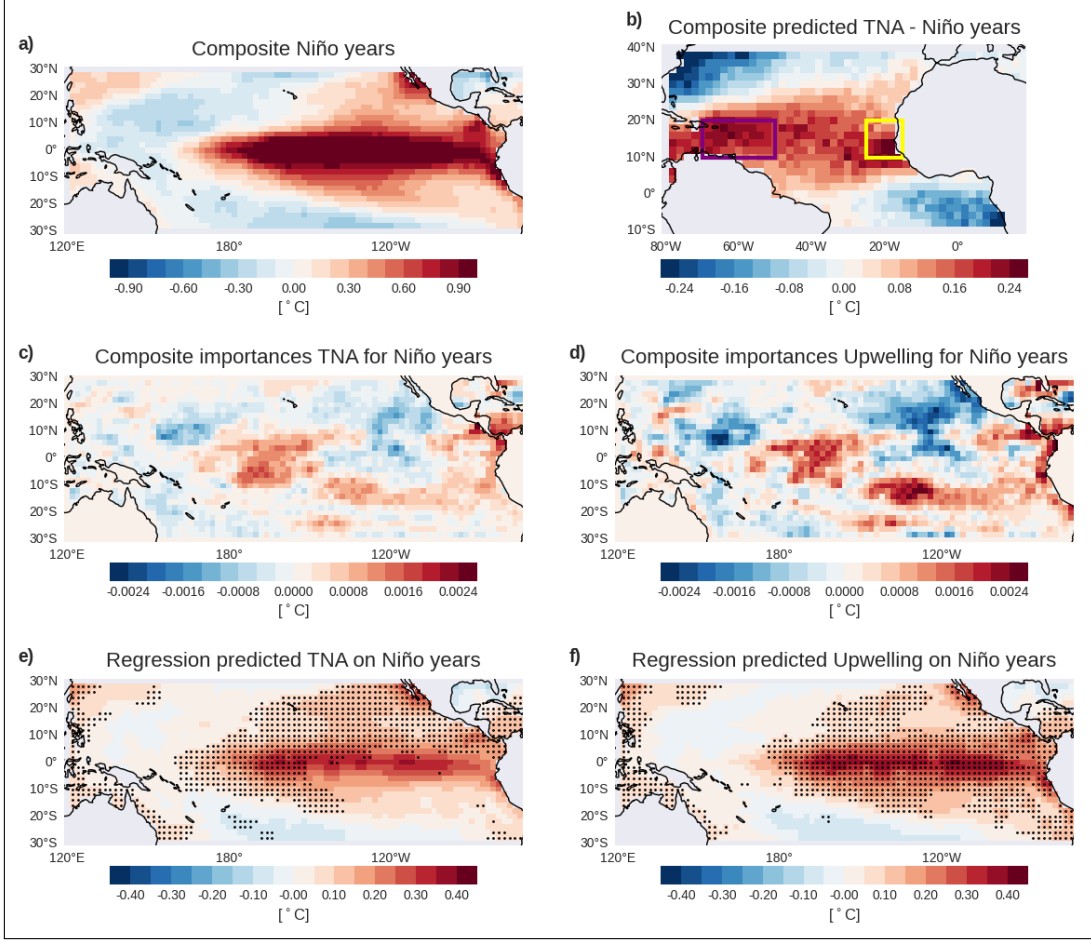

**Figure 3.** Composite of model predictions corresponding to Niño events. Panels show: (a) observed Pacific SST anomalies during DJF; (b) predicted Atlantic SST anomalies during MAM; (c) attribution maps over the predictor field related to the western Tropical North Atlantic (WTNA) region predictions (purple box shown in panel b); (d) attribution maps over the predictor field related to the upwelling region predictions (yellow box shown in panel b); (e) regression of the TNA index on Pacific DJF SST; and (f) regression of the upwelling index on Pacific DJF SST. The stippling in panels (e) and (f) indicates regions where the regression is significant at the 95% confidence level, according to a Monte Carlo two-tailed significance test. Attribution maps (c) and (d) represent the relative contribution of each grid point in the predictor field to the forecasted value in the target region. Note that the sum of the attribution values within each map equals the predicted anomaly in the corresponding region (i.e., the sum of values in panel c matches the WTNA anomaly within the purple box in panel b).

The metrics of model performance across the entire hindcast period are summarized in the output figure of the function (Fig. 2). High skill is shown in the whole north tropical Atlantic region, with maximum values (up to 0.9) over the western side of the basin (Fig. 2a). Notably, the skill in the TNA region remains positive and ranging from 0.4 to 0.8 in most of the years, with improved performance during certain decades, as during the 1980s (Fig. 2b). In terms of the root mean square error (RMSE), Figure 2c) highlights three regions where model errors are more pronounced: the Atlantic Niño, the Gulf Stream, and the




MSCU system. These areas coincide with regions exhibiting the highest SST variability. Interestingly, it is worth emphasizing that the model achieves a high level of skill with only tropical Pacific information, allowing for the quantification of the impact of boreal winter ENSO conditions on the tropical Atlantic state in the subsequent spring.

To better understand the model behaviour, we have explored the SST signal in the tropical Pacific (predictor) and TNA (predictand) during el Niño events. In particular, the attributions of the predicted SST signal in WTNA and SMSCU regions during el Niño events have been computed and are presented as composite maps in Figure 3. The predictor field corresponds to a canonical El Niño pattern (Fig.3a), while the predicted response of the model shows a warming of the TNA, along with a cooling in both the Niño Atlantic region and the Gulf Stream area (Fig.3b). Notably, the mean attributions for predicting both the WTNA and the SMSCU indices exhibit similar spatial structures, differing mainly in the magnitude of their contributions,
being higher for the MSCU region (Fig.3c-d). In particular, the central Pacific and the eastern equatorial Pacific, next to South American coasts, appear to positively contribute to the SST anomalies observed in both regions of interest. Additionally, two areas in the western north tropical Pacific and near Californian coast emerge as negative contributors to the WTNA and SMSCU warmings. To compare with traditional linear statistical approaches, we have added the regression of the MAM WTNA and SMSCU predicted indices over the Pacific DJF SST anomalies during Niño events (Fig.3e-f). Comparing both methods
highlights the added value of the model-based attributions. Not only do they identify a significant contribution from a central Pacific region and the coastal area of the western Pacific basin to the overall signal, but they also emphasize other regions that act oppositely, effectively contributing to the cooling of the study areas. These key areas over the central Pacific are consistent with findings from recent studies (Wade et al., 2023), whereas the regions of cooling remain open questions that could be further explored in future studies through sensitivity experiments. Model predictions and their attributions for two specific
samples – 1986 and 1997 – as well as the composite for La Niña events, can be found in the supplementary material (Fig. S1-S3 in the Supplementary Material).

## 4.2    Modelling north and tropical Pacific - European precipitation teleconnection

In this section, we focus on the relationship at the seasonal scale between north and tropical Pacific SST anomalies and European precipitation during boreal autumn (October–November–December; OND), a challenging target variable for dynamical
models (Johnson et al., 2019). In the North Atlantic, the dominant mode of atmospheric variability is the North Atlantic Oscillation (NAO), which modulates the pressure gradient between the Icelandic Low and the Azores High (Rogers, 1997; Trigo et al., 2002). This gradient, in turn, determines the storm tracks over Europe and consequently influences precipitation. Although the NAO signal largely arises from internal variability, external factors such as ENSO can modulate its centers of action via tropospheric and stratospheric pathways (Rodwell et al., 1999; Rodríguez-Fonseca et al., 2016). The interplay with other
climate modes, including the Atlantic Multidecadal Oscillation (AMO) and the Pacific Decadal Oscillation (PDO), contributes to the non-stationarity of this teleconnection and, consequently, to the varying influence of ENSO on European precipitation over time (López-Parages and Rodríguez-Fonseca, 2012).

For this analysis, SST data from the HadISST dataset were used as the predictor field (Rayner et al., 2003), while precipitation data from the Climatic Research Unit gridded Time Series (CRU TS) dataset served as the predictand (Harris et al.,





| Parameter | Value | Parameter | Value |
|---|---|---|---|
| **Input/Output, Temporal and Spatial Settings** | | | |
| region_predictor | Pacific | time_lims | 1901–2019 |
| lat_lims_x | 75, -20 | lon_lims_x | 120, 300 |
| lat_lims_y | 75, 35 | lon_lims_y | -10, 30 |
| **Preprocessing** | | | |
| detrend_x/y | True/True | detrend_x/y_window | 30/30 |
| mean_seasonal_method_x/y | True/False | scale_x/y | 1/1 |
| regrid_degree_x/y | 2/1 | months_x | 10, 11, 12 |
| months_y | 10, 11, 12 | months_skip_x/y | None/None |
| **Model Architecture** | | | |
| layer_sizes | 1021, 256, 64, 256, 1024 | activations | ELU |
| num_conv_layers | 0 | dropout_rates | 0.1 |
| use_dropout | True | use_batch_norm | True |
| use_initializer | True | kernel_regularizer | l2 |
| use_init_skip_connections | False | use_inter_skip_connections | False |
| **Training Settings** | | | |
| learning_rate | 0.0001 | epochs | 2500 |
| p_value | 0.1 | jump_year | 0 |

**Table 3.** Table summarizing the model hyperparameters and preprocessing settings used in the simulation of Pacific SSTs and European precipitation in OND.

2020). Additionally, geopotential height at 200 hPa (Z200) from ERA-20C reanalysis is used for dynamical analysis (Poli et al., 2016). The predictor region covers the Pacific basin [20°S–75°N; 120°E–70°W] to capture the ENSO and extratropical Pacific related variability, while the predictand region is defined by the European continent [35°N–75°N; 10°W–30°E]. The hyperparameters and other modelling parameters are detailed in Table 3. Unlike the previous case study, precipitation was aggregated from monthly data (by setting the parameter $mean\_seasonal\_method\_y = False$), and the preprocessing steps are outlined in Listing 2.

After performing a leave-one-out cross-validation, the results are summarized in Figure 4. The ACC map indicates that only northern Scandinavia and eastern central Europe exhibit significant positive skill (Fig. 4a). The RMSE map indicates higher errors in regions with greater precipitation variability, such as Norway, the United Kingdom, and northwestern Spain (Fig. 4c). At first glance, these results might suggest that either the model fails to capture the teleconnection or that the Pacific SST and European precipitation link is inherently non-stationary. The temporal evolution of the ACC and RMSE (Fig. 4b-d) supports the latter hypothesis; there are periods such as the 1950s-1970s, when the model displays considerable skill (ACC > 0.4), contrasted with periods like the 2000s where skill is minimal (ACC ≈ 0).





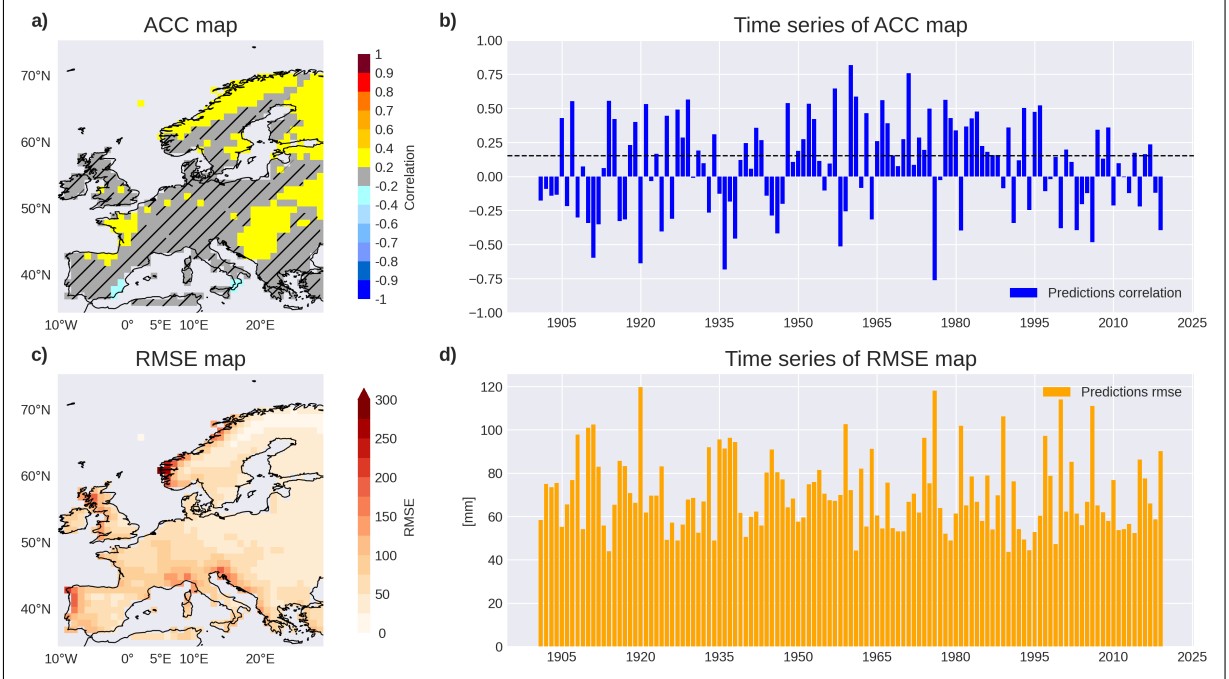

**Figure 4.** Predictability of European precipitation variability from north and tropical Pacific SST anomalies. Panel showing model performance metrics over the full period (1901–2019) using a leave-one-out cross-validation approach for predicting the precipitation anomaly field over Europe during OND, with Pacific SST from OND as the predictor. Predictions are compared against observed OND precipitation anomalies. Specifically: (a) ACC spatial map, correlating for each grid point the observed and predicted time series; (b) Time series of ACC maps, correlating for each year the observed and predicted spatial patterns; (c) RMSE spatial map; and (d) Time series of RMSE maps, computed analogously to the ACC metrics, all calculated between predicted and observed fields. The ACC (RMSE) time series show the correlation (error) between predicted and observed global mean SST anomalies over time. Statistically significant results, based on a one-tailed t-test with a significance level defined in the hyperparameters (95%), are indicated by the non-dashed regions in panel (a) and values above the dashed line in panel (b).

To gain further insight into the model predictions and to understand the type of European precipitation variability it is capturing, we conduct additional analyses. Concretely, we employ the built-in `PC-analysis` functionality of NN4CAST to
extract EOFs. The leading mode of variability of the OND precipitation fields—explaining 30% and 45% of the total variance in the observational datasets, respectively—exhibits a coherent spatial pattern that the model reproduces with remarkable accuracy (Fig.5a-b). In particular, the model effectively captures the core features of this mode, especially in regions where this mode of variability has the higher impacts, such as Norway, the United Kingdom, western Europe, and the Balkans. While a slight underestimation in the overall amplitude is evident, the spatial correlation between the simulated and observed patterns
reaches r = 0.98, indicating that the model skilfully represents the structure and geographical distribution of the dominant precipitation variability during this season.





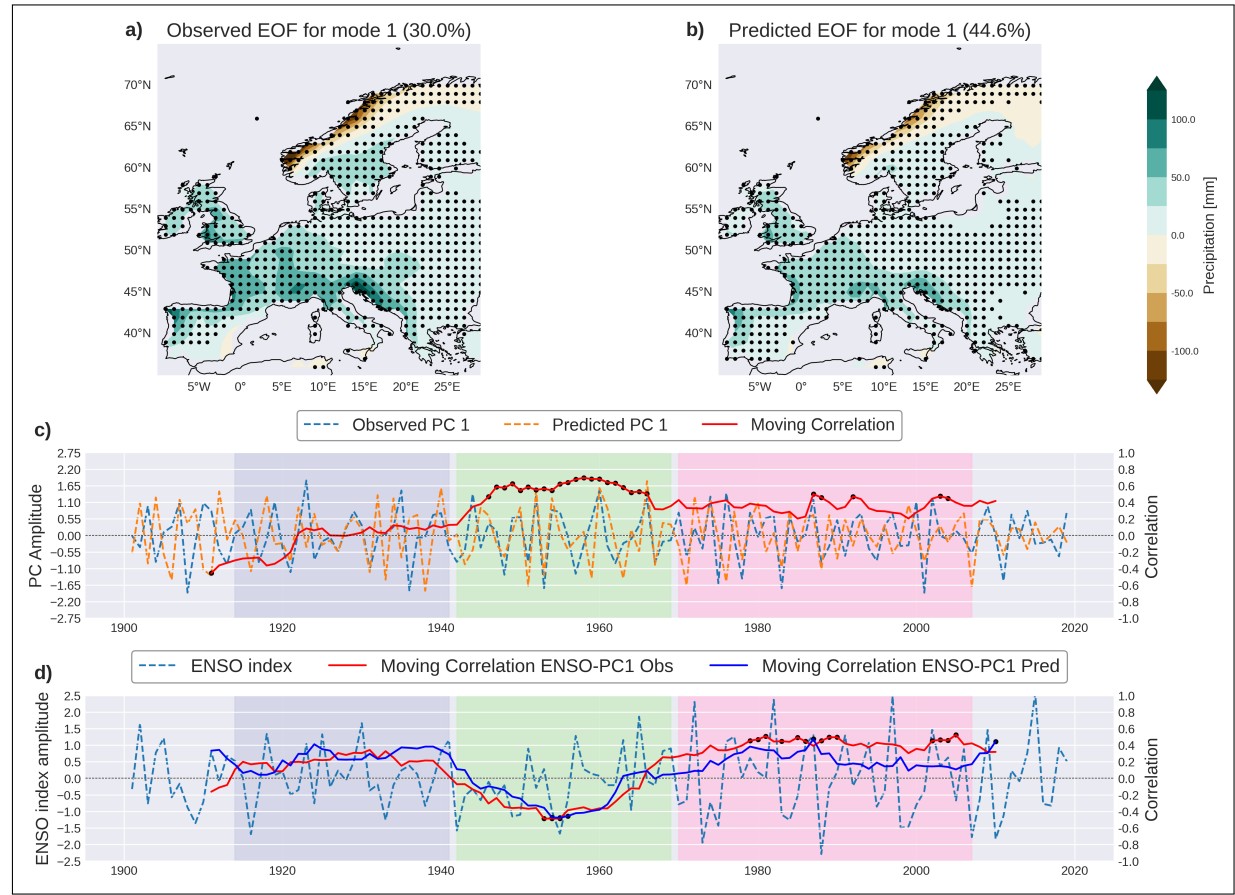

**Figure 5.** Principal Component Analysis (PCA) of precipitation anomalies over Europe during OND, for both observed and predicted datasets. Panels (a) and (b) show the spatial pattern of the first mode of variability for the observed and predicted fields, respectively, with the percentage of explained variance indicated in parentheses. Statistical significance in panels (a) and (b) is assessed using a Monte Carlo test at the 95% confidence level. Panel (c) displays the temporal evolution of the principal component (PC) associated with the first mode, representing its high-frequency component after applying a Butterworth filter with a 7-year cut-off frequency, along with the 20-year centered moving window correlation between the observed and predicted PC time series. Panel (d) presents the Niño 3.4 index time series and its 20-year centered moving correlations with both the observed and predicted PCs. Data points in panels (c) and (d) where the moving window correlation is statistically significant at the 95% confidence level, based on a two-tailed t-test, are highlighted. Three shaded periods indicate intervals selected for further analysis following (López-Parages and Rodríguez-Fonseca, 2012).

To better understand how this variability evolves over time, the associated PC time series were first temporally filtered to retain only their high-frequency components by applying a Butterworth filter with a 7-year cutoff frequency. Twenty-year centred moving-window correlations between the observed and predicted PCs (Fig.5c) reveal that model skill is highest during 1940–1969 (r≈0.4–0.7), drops to near zero during 1910–1940, and remains modest (r≈0.2–0.5) over 1970–2010. To identify the climatic driver behind these fluctuations, we then calculated the moving-window correlations between each high-frequency






PC and the Niño 3.4 index (Fig.5d). During the 1940–1969 period of higher skill, the PC–Niño 3.4 correlation is strongly negative, indicating that La Niña events coincide with increased precipitation over western Europe. In contrast, the periods of reduced skill (1910–1940 and 1970–2010) are characterized by weak to positive PC–Niño 3.4 correlations, suggesting a
weakening or reversal of the ENSO teleconnection. These results demonstrate that the model not only reproduces the dominant precipitation variability but also captures the non-stationary nature of the ENSO-European precipitation.

To further investigate the non-stationary ENSO–European precipitation relationship and compare our results with previous studies (López-Parages and Rodríguez-Fonseca, 2012), we defined three intervals according to the model skill in reproducing the PC of the leading mode of European precipitation variability (Fig.5c): the low-skill periods 1914–1941 and 1970–2007,
and the high-skill period 1942–1969. The corresponding ACC maps for each period are presented in Figure 6. In the maps corresponding to the low-skill periods (Figs. 6a and 6c), only a few small regions in northeastern Europe show significantly positive skill. In contrast, during the 1942-1969 period (Fig. 6b), statistically significant skill is shown in most of the European continent. This temporal evolution of model performance highlights a marked non-stationarity in the ENSO–European precipitation teleconnection, motivating a closer examination of the associated large-scale dynamical conditions. To this end, and with the aim of identifying potential drivers of the observed variations in skill, we analyzed the first predicted principal component
of European precipitation (shown in orange in Fig.5c). This index was used to compute regression maps of PC1 during OND over simultaneous anomalous SST and Z200 fields for each of the three periods. Specifically, during periods 1914-1941 and 1970-2007 (Figs. 6b and 6f), the SST pattern suggests a signal in the Pacific, indicating a relationship between central Pacific SST positive anomalies and increased precipitation in western Europe (shaded in Fig. 5c-f). Regarding the Z200 regressions, a
Gill-type response characteristic of ENSO events is evident in the Pacific, including its modulation of the Aleutian Low; however, no statistically significant dynamic pattern emerges over the North Atlantic that might influence European precipitation (contours in Fig. 5c-f).

In contrast, during the 1942–1969 interval, the teleconnection changes such that La Niña conditions correspond to enhanced precipitation over western Europe (shaded in Fig. 5d). Moreover, significant regression signals emerge in several extratropical
basins, such as the coastal region of California, suggesting that these areas may play an important role in modulating the teleconnection during this period. As for the regression on Z200, a pattern of circumpolar wave trains is evident, possibly originating from the western Pacific region (contours in Fig.6e). To verify this hypothesis, sensitivity experiments should be conducted to analyse the impact that this region may have on European precipitation.

Nonetheless, this non-stationarity analysis aligns with the study by (López-Parages and Rodríguez-Fonseca, 2012), which
reported a stronger relationship between ENSO and European precipitation during the period 1942–1969, coinciding with the period of highest skill of the model. This analysis has also been performed on the first observed principal component of the precipitation, yielding similar results in the SST regressions in particular, suggesting that the model has successfully captured the associated teleconnection (Fig. S4 in Supplementary Material).



**Figure 6.** Panel of correlation maps between predicted and observed precipitation anomalies over Europe during OND for three distinct periods: (a) 1914–1941, (b) 1942–1969, and (c) 1970–2007. Data points indicate grid cells where the correlation is statistically significant at the 95% confidence level based on a t-test. The panel also includes regression maps of the first predicted principal component (shown in orange in Fig. 5c) onto the SST (shaded) and 200 hPa geopotential height (Z200; contour) fields during OND. Statistical significance of the regressions is indicated by stippling over the SST maps and black contour lines over the Z200 maps, according to a Monte Carlo two-tailed test at the 95% confidence level.





## 5 Conclusions

Seasonal predictions play a crucial role in climate services, offering valuable insights for sectors such as water resource man­agement, energy demand forecasting and agricultural planning, among others. Earth system models make these predictions by taking into account the interactions of the different components of the climate system. Particularly in ocean-atmosphere interactions, the scale of ocean adjustment is slower, which gives predictability to the impacts on the atmosphere. However, they are subject to different challenges, not only due to the misrepresentation of certain physical processes due to the limited

spatio-temporal resolution, but also due to errors in the initial and boundary conditions. An alternative emerges in the statistical approaches trained with observations for performing these simulations. Nevertheless, these statistical approaches ((linear re­gression, maximum covariance analysis, etc) focus on the relationships between a predictor and a predictand within a specified time lag between them, avoiding the problem of the numerical solving of the dynamical equations of the atmosphere. Neverthe­less, their main drawbacks rely on the limited number of past cases, the non-stationarity behaviour of certain relationships and

the assumption of a linear relationship between the fields. For this reason, we developed NN4CAST, a tool designed to assess the sensitivity of a target impact climate variable to variations in drivers operating at seasonal time scales, such as changes in SST anomalies across different regions. One of the key advantages of this modelling framework is its versatility and ease of use. It does not require users to have extensive knowledge of deep learning concepts and techniques, making it more accessible for domain experts to apply deep learning methods in their respective fields of study. Its primary applications include:

· Providing a versatile and user-friendly tool for ease of implementation and application.

· Modelling relationships between fields, including non-linear components, with performance evaluated using various skill and error metrics.

· Assessing potential predictors and exploring the impact of target variables on specific drivers to enhance predictability.

· Providing a versatile and user-friendly tool for ease of implementation and application.

· Identifying windows of opportunity where relationships between fields exhibit stronger predictability.

· Facilitating the analysis of changes in predictability by examining the model attributions in the predictions.

Two case studies, based on well-known climate teleconnections,have been selected to analyse the potential applications of the proposed tool. First, the teleconnection between ENSO in DJF and the tropical North Atlantic in MAM serves as a bench­mark of a well-known and robust teleconnection, allowing for an evaluation of the model skill and the identification of the

main sources of predictability through attribution techniques. Second, the teleconnection between the north and tropical Pa­cific and concurrent European precipitation in OND highlights the added value of the tool when dealing with a more complex and non-stationary relationship. In this context, the model enables the identification of periods of enhanced predictability and provides insights into the drivers of such variability. These complementary approaches offer valuable contributions to the sci­entific community and support the improvement of current seasonal forecasting systems. In our first case study, examining the



ENSO–TNA SST teleconnection, the model successfully reproduces the canonical relationship with a high skill and, through attribution analysis, identifies the central Pacific as a key region of influence that traditional regression methods fail to detect. In the second case study, assessing Pacific SST impacts on European precipitation, the model accurately captures the teleconnection non-stationarity behaviour, obtaining the highest skill during the period of negative ENSO–European precipitation correlation and reproducing the dynamical patterns characteristic of that period.

The case studies presented demonstrate that the NN4CAST framework enables model assessment via metrics that identify windows of opportunity, complemented by a comprehensive visual output. Model skill metrics and the analysis of attributions in the predictive field, outcomes of NN4CAST, will allow users to better understand climate teleconnections, its underlying physical mechanisms and to identify the most relevant areas for the model prediction. The NN4CAST framework further facilitates the implementation of pseudo-sensitivity experiments, whereby users can, for instance, select different SST regions as predictors and evaluate their individual and combined contributions by applying them both jointly and separately. This capability, together with the possibility to vary predictands, employ different datasets, and introduce noise, enhances the framework utility for systematically assessing model sensitivities and robustness. These capabilities significantly enhance its versatility for exploring and understanding climate predictability. Additionally, this framework is designed to allow the integration of more complex deep learning architectures currently used in meteorological modelling, such as transformers. This capability enables a direct evaluation of the strengths and limitations of each architecture within the modeling framework. Furthermore, plans are underway to extend the NN4CAST framework to additional applications relevant to the scientific community, including integration with tools such as ESMValTool, thereby enhancing its utility for comprehensive climate model evaluation and diagnostics.

*Code and data availability.* The current version of NN4CAST is available from the Gihub repository at https://github.com/Victorgf00/nn4cast, under the MIT licence. The exact version of the model used to produce the results used in this paper is archived on Zenodo at https://doi.org/10.5281/zenodo.14011998 (Galván Fraile et al., 2024). The scripts to run the model and produce the plots used in this paper as well as the input data are also archived on Zenodo at https://doi.org/10.5281/zenodo.15682872 (Galván Fraile et al., 2025).

*Author contributions.* The paper was written by VG, with contributions and suggestions from all the authors. The code was developed by VG.

*Competing interests.* The contact author has declared that none of the authors has any competing interests.



ther geographical representation in this paper. While Copernicus Publications makes every effort to include appropriate place names, the final responsibility lies with the authors.

*Financial support.* This research has been supported by the Spanish Ministry of Science, Innovation and Universities through the National
420 Program FPU (grant no. AP-2022-02162) and the Oceans for Future project (Innovative climate services using ocean information and communication with society. grant no. TED2021-130106B-I00 funded by MCIN/AEI /10.13039/501100011033 and by the European Union Next GenerationEU/ PRTR Strategic Projects oriented to the Ecological Transition and the Digital Transition. Call 2021). MMR has been supported by Ramón y Cajal (RYC2022-038454-I, funded by MCIN/AEI/10.13039/501100011033 and co-funded by the FSE+, European Union)



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
