# Peer review of "Assessing seasonal climate predictability using a deep learning application: NN4CAST"

_EGUsphere, 2025_

## Author Comment (AC1)

**Response to Reviewer 1**

October 20, 2025

**Detailed Comments**

Responses are marked in blue.

**General Comments:**

The authors provide a tool that may be utilized to research seasonal predictability using basic deep learning methods. The code library provides a pipeline to preprocess data, train the model, evaluate, and calculate some metrics/attributions, based on a user-defined namelist and input files. Although the model will not achieve state-of-the-art skill, it does have potential for mechanistic studies through explainable AI. However, I do not believe the manuscript in its current state effectively communicates this message.

1. The analysis of the teleconnection between DJF Pacific tropical SST and MAM tropical Atlantic SST and related evaluation of the model is not valid, due to the region of the input predictor field, which includes parts of the western tropical Atlantic. Looking through the individual Integrated\*Gradient attribution samples on Zenodo, it is clear that the largest attributions are most often in this area, rather than in the tropical Pacific. This is also confirmed by calculating correlations between areal-averaged SST in the target WTNA or SMSCU region with the input SST field. This leads to unrealistic, inflated skill in Figure 2, which is a result of the inclusion of the west Atlantic in the input fields, rather than the Pacific-Atantic teleconnection, as stated in the text (line 266-267).

We thank the reviewer for this insightful comment. We agree that including the small region of the Caribbean in the predictor field could introduce information that is more directly related to the persistence of the predictand. To assess the impact of this effect, we performed an additional experiment in which the Caribbean region has been masked in the predictor field. In addition, a complementary mask has been also applied to the predictand field to exclude areas in the Pacific. This allows us to evaluate the model skill over the Caribbean without any overlap between predictor and predictand domains.

Importantly, this adjustment did not require modifications to the core model code, as the masking can be applied directly at the preprocessing stage of the SST datasets before they are introduced into the model pipeline.

The results of this experiment show that the model skill is slightly reduced in certain areas around 20°N, such as the Gulf of Mexico, but maintains the spatial structure with correlation scores over 0.6-0.7 and similar RMSE (Figure R1) than in the original experiment.

In the revised manuscript we have updated figure 2 to include the results of this new experiment instead of the original experiment. In addition, the outputs of the model on

Zenodo have also been updated to incorporate these new simulations, ensuring full reproducibility of the results.

Figure R1. Predictability of tropical North Atlantic SST variability from tropical Pacific anomalies. Panels showing model performance metrics over the full period (1901–2019) using a leave-one-out cross-validation approach for predicting the SST anomaly field of the tropical North Atlantic during MAM, with Tropical Pacific SST from DJF as the predictor. The predictions are compared against observed MAM SST anomalies. Specifically: (a) ACC spatial map, correlating at each grid point the observed and predicted time series (temporal dimension); (b) Time series of ACC maps, correlating for each year the observed and predicted spatial patterns (spatial dimension); (c) RMSE spatial map, computed analogously to (a); and (d) Time series of RMSE maps, computed analogously to (b), all calculated between predicted and observed fields. The ACC (RMSE) time series show the correlation (error) between predicted and observed global mean SST anomalies over time. Statistically significant results, determined using a one-tailed t-test at the 95\% significance level, are indicated by the non-dashed regions in panel (a) and values above the dashed line in panel (b).

2. The discussion surrounding XAI in Figure 3 is unconvincing. Although the model attribution plot (Fig 3c) shows more spatial variability than the simple regression (Fig 3e), this does not necessarily mean there is added value. The work would benefit from further exploring the physical mechanisms associated with the Integrated Gradients attribution. There is not a clear connection between the spatial variance in Fig 3c and the citation of Wade et al. 2023 in the text.

We are very grateful for these detailed and constructive comments. They have prompted us to deepen our analysis and to clarify important points regarding mechanisms, robustness, and sample dependence.

First, concerning Wade et al. (2023): their study shows that SST variability in the Senegalese upwelling is connected to Pacific ENSO anomalies, particularly during 1960–1990 (their Figure 6). This coincides with our results (Figure R1b), where the model skill is higher in those decades, and with anomalies in the Pacific having maximum values on the central Pacific when the model predicts a warming in the region of the upwelling Figure R4d).

In addition, attribution results reveal that areas located in the central (180°W–150°W) and easternmost equatorial Pacific (110°W–90°W) significantly contribute to SST anomalies over the SMSCU (Fig. 3c). The reference to Wade et al. (2023) is made precisely because they identify this equatorial Pacific signal as a key remote driver of Senegalese SST variability (see their Fig. 6). In our case, NN4CAST shows significant skill throughout the century, with higher scores during 1960-1990 (Fig. 2b), coinciding with the period in which Wade et al. found a strong relation between coastal upwelling and central Pacific SSTs (their Fig. 3). The Integrated Gradients attribution, confined mainly to this region, therefore confirms the central equatorial Pacific as a key remote driver of SST variability in both WTNA and SMSCU. This has been clarified in the new version of the manuscript (Lines 272-280).

Wade, M., Rodríguez-Fonseca, B., Martín-Rey, M., Lazar, A., López-Parages, J., & Gaye, A. T. (2023). Interdecadal changes in SST variability drivers in the Senegalese-upwelling: the impact of ENSO. *Climate Dynamics*, *60*(3), 667-685.

What is the sample size? There is only a ~100 year record that is being used, with even fewer El Niño's, so I am skeptical of the robustness of model attribution. How much does the attribution pattern change with different initial seeds?

Regarding the sample size: our training period covers approximately 120 years (1901–2019). Within this span, there are on the order of 35-40 El Niño and 45-50 La Niña events, meaning that the strongest events remain relatively few. However, 100–120 years of observational record is generally considered sufficient for studies of climate variability (e.g., Trenberth (1997); Ray & Giese (2012)). Furthermore, the NN4CAST model has been tested against observations to reproduce known teleconnection patterns, providing confidence in the robustness of the attribution results. In the revised manuscript, the composites are constructed based on the WTNA and SMSCU indices rather than directly on ENSO events, which reduces dependence on the relatively few strongest ENSO events.

Concerning variation with initial seeds: we have now conducted ten simulations in which the only difference is the random seed (which we have made explicit as a hyperparameter in the library; previously it was implicit). Thus, seed variability does not appear to compromise the robustness of the main attribution findings. To illustrate the spread in the results according to initial seeds, we have computed the longitudinal and latitudinal averages of importances across the predictor field for the positive events of WNTA for the different initializations (see Figure R2), as well as the spatial average of the importance for those positive events across the model initializations (see Figure R3). The results show that while small-scale details vary somewhat across seeds, the large-scale attribution patterns remain stable. In particular, although there is some variability in the sign of attributions in certain regions, in the areas that the models assign the highest importance for their predictions, the attributions are consistent across all

initializations. Notably, the central Pacific consistently emerges as an important region (Figure R3), in agreement with the previous results (Figure 3).

**Figure R2.** Mean longitudinal and latitudinal distributions of variable importance for predicting positive WTNA events. The values represent averages across 10 independent model initializations obtained by varying the random seeds.

**Figure R3.** Spatial distribution of the mean importance of each predictor for forecasting positive WTNA events. The values are obtained by averaging across multiple model initializations with different random seeds. Markers indicate the level of agreement in sign among models: '\*' for less than 50% agreement, '.' for 50–75% agreement, and no marker for more than 75% agreement.

Trenberth, K. E. (1997). The definition of el niño. *Bulletin of the American Meteorological Society*, 78(12), 2771-2778.

Ray, S., & Giese, B. S. (2012). Historical changes in El Niño and La Niña characteristics in an ocean reanalysis. *Journal of Geophysical Research: Oceans*, *117*(C11).

Have you tried calculating attribution plots, compositing on a warm WTNA or SMSCU, rather than ENSO?

Following your suggestion, we have computed composite maps based on the predicted Atlantic indices (WTNA and SMSCU) in addition to the ENSO-based composites originally reported. These new composites are presented in Figure R4. In the first column of Figure R4 we show composites of the model SST predictions for both indices (panels a and c) together with surface wind anomalies for those events in MAM, to examine local dynamical changes that may underpin the mechanism. For example, SMSCU-positive composites reveal a local strengthening of southwesterly winds, blowing along the Senegalese coast, which can contribute to a reduction of the coastal upwelling and strong coastal SST warming (Figure R4d). In contrast, the WTNA box shows weaker wind anomalies close to the African coast, consistent with the weaker predicted SST signal (Figure R4a). To better understand which regions contribute to the SST signals in each index, we show both the predictor-field composites and the attribution maps. For both WTNA and SMSCU, the central Pacific emerges as an important region in the attribution maps, specifically around 170°E-150°W, indicating that the model often leverages Pacific-centered anomalies when predicting these Atlantic indices (Figure R4b,d).

To understand the physical mechanism and atmospheric pathways and to corroborate the robustness of the relation found with ENSO, we have generated two additional composites of anomalous surface wind, mean sea level pressure (SLP), and geopotential height at 200 hPa (Z200) (Figure R4, panels e and f). In these fields, the region highlighted by the model in the central Pacific corresponds to an area of pronounced wind convergence and an Gill atmospheric response to an equatorial warming (Gill, 1980), which is characterized by 2 symmetric anticyclones at both sides of the equator in upper levels This tropical atmospheric response is part of a broader wave response, which propagates to the extratropics towards the Atlantic as an extratropical Rossby-wave, producing a negative NAO like pattern over the North Atlantic (more clear for WNTA events). This associated weakening of the subtropical high pressure system during a negative NAO weakens the trade winds over the TNA region (Figure R4g,h). This physical mechanism linking central Pacific SST anomalies to the tropical North Atlantic indices is in accordance with the literature (Horel & Wallace (1981); Czaja et al. (2002)). We have also computed analogous composites for negative phases of WTNA and SMSCU with consistent results. These are included in the revised manuscript for completeness. In addition to the extratropical Rossby wave, a secondary Gill response also appears over the equatorial Atlantic, as a result of the anomalous upper level convergence from the anomalous Walker circulation. This signal, which is baroclinic, also contributes to the weakening of the trades and upwelling, in agreement with García-Serrano et al (2017). The difference between WTAN and MSCU is the extension of this Gill response, which is more regional for the MSCU

These WTNA/SMSCU-based composites are complementary to the ENSO-based analysis shown in the previous version of the manuscript. ENSO-conditioned composites provide information about the model performance of the Pacific-Atlantic teleconnection

specifically under ENSO events, whereas the Atlantic-index based composites show which remote features the model exploits to predict Atlantic indices independently of ENSO. To maintain clarity of presentation, we have focused on the WTNA/SMSCU composites in the main text (Figure R4). We agree with the referee that the index-based composites provide valuable information about the key predictor regions of SST for WTNA and SMSCU. In the revised manuscript, the ENSO-based composites are presented in the Supplementary Material, while selected WTNA/SMSCU composites are shown in Figure 3, allowing readers to compare ENSO-conditioned and Atlantic-index–conditioned attribution results.

**Figure R4.** Composites of model anomalous SST predictions, predictor fields, and attribution maps for positive predicted WTNA and SMSCU, based on 28 and 26 events, respectively, during the period 1901-2019. Panels a) and c) show the predicted mean SST anomalies in the Atlantic during MAM together with surface wind anomalies indicated by arrows. Panels b) and d) show the attribution maps over the predictor fields with SST in contours and surface winds in arrows. Panels e) and f) display global composites of MAM anomalies in sea level pressure (shading), 200 hPa geopotential height (contours), and surface winds for positive WTNA and SMSCU events. Attribution maps indicate the relative contribution of each grid point in the predictor field to the forecasted value in the target region, with the sum of the values within each map matching the predicted anomaly in the corresponding index region (i.e., the sum of values in panel c matches the WTNA anomaly within the purple box in panel a).

Gill, A. E. (1980). Some simple solutions for heat-induced tropical circulation. *Quarterly Journal of the Royal Meteorological Society*, *106*(449), 447-462.

Horel, J. D., & Wallace, J. M. (1981). Planetary-scale atmospheric phenomena associated with the Southern Oscillation. *Monthly Weather Review*, *109*(4), 813-829.

Czaja, A., Van der Vaart, P., & Marshall, J. (2002). A diagnostic study of the role of remote forcing in tropical Atlantic variability. *Journal of Climate*, *15*(22), 3280-3290.

García-Serrano, J., Cassou, C., Douville, H., Giannini, A., & Doblas-Reyes, F. J. (2017). Revisiting the ENSO teleconnection to the tropical North Atlantic. Journal of Climate, 30(17), 6945-6957.

3. The analysis of European precipitation is useful for showing how the predictability varies between different periods. However, the regression analysis in Figure 6 is a little confusing, as you could perform the exact same regression with only observational data, yielding more faithful results and yielding the same conclusion regarding ENSO and European precipitation. Figure 5 shows the model can reproduce some of the same trends as observations, but doesn't reveal any new insights not available from solely observations.

We thank the reviewer for this comment. The primary goal here is to assess whether the model can reproduce the variability of European precipitation and its decadal changes, including its modulation by ENSO impact. To this end, comparing the regression using model predictions with the regression using observations serves as a consistency check: it validates the model ability to capture this teleconnection and its temporal evolution (non-stationarity behavior). Importantly, the purpose of this figure is not to provide a new observational analysis, but to demonstrate that the model itself reliably reproduces these patterns under the leave-one-out cross-validation framework.

Similarly to the previous analysis, it does not seem like the model is directly capturing a connection between ENSO and European precipitation, based on the individual attribution plots on Zenodo, which mostly show the model thinks SST anomalies in the extratropical Pacific and Atlantic Ocean are important. What could maybe be useful is to look at the attribution plots for precipitation in skillful regions during 1942-1969? Maybe there is a change in the background state (e.g. the extratropical jet), which changes the

propagation of the extratropical Rossby wavetrains that affect European precipitation and thus predictability?

We thank the reviewer for this helpful comment. Following the suggestion, we analyzed composites of positive and negative events based on a predicted precipitation index over western–central Europe (purple box in Fig. R5a), where the model shows significant skill (Figure 6 in the previous version of the manuscript) for the periods P2 (1942–1969) and P3 (1970–2007) (figure R5). On this basis, we first show composites of European predicted precipitation anomalies (Figs. R5a,d and R6a,d), along with global anomalies of wind (U10, V10), geopotential height (Z200), SST, and precipitation (Fig. R5c,f and R6c,f) to analyze the teleconnection mechanisms associated with the anomalous rainfall predicted by the model. We also present the attribution maps (Fig. R5b,e and Fig. R6b,e) to assess the regions contributing to the European precipitation signal.

The results reveal distinct mechanisms between the two periods. During P2 (1942–1967), a weak La Niña appears to induce enhanced convection and positive precipitation anomalies over the Maritime Continent, which act as a source of Rossby waves and generate a Gill-type response in the upper troposphere. The circulation anomalies suggest the presence of two Rossby wave trains, one propagating westward from the eastern Pacific and another emanating from the Maritime Continent region in association with the tropical precipitation anomalies (Fig. R5c). In contrast, during P3, a strong El Niño event dominates, with a clearer Gill-type response that generates an atmospheric extratropical Rossby wave train propagating into the extratropics (Figure R5f). An analogous analysis for negative precipitation anomalies yields an approximately opposite mechanism (see Fig. R6).

The differences in the mechanisms driving anomalous European precipitation between these two periods could be related to changes in the background state (Figure R7). For example, Fig. R7b shows that P2 is characterized by a weaker meridional SST gradient in both the Pacific and the Atlantic compared to P3, resulting in a weakened and southward-displaced extratropical jet. These climatological changes could explain the differences in the teleconnection patterns observed in Fig. R5: in P2, the Pacific-Europe link is relatively weak, with the apparent Rossby wave source located over the Indochina region (due to anomalous convection), whereas in P3, a stronger meridional gradient allows a clear tropical Pacific wave source, consistent with a Gill-type response, to influence European precipitation

The attribution maps allow us to clarify the regions contributing to the European precipitation signal. In P2, the maps indicate that most of the predictive signal comes from an extratropical region around 40°N and 160°E, reflecting the weakening of the Aleutian Low. In P3, in addition to this extratropical region, a tropical Pacific contribution emerges. As noted previously, to specifically assess the predictability arising from SST anomalies alone (rather than from dynamical factors that are indirectly reflected in the SST field), the simulations could be repeated with an increased lag between the predictor and predictand fields. This approach would allow a clearer separation of the SST-driven signal from atmospheric circulation effects.

In the revised manuscript, this analysis has been clarified by not only analyzing the dynamics of the mechanisms and its relation with the importances of the model, but also highlighting the impact of the changes of the mean state in the teleconnection mechanism (Lines 335-366).

Figure R5. Composites of model anomalous precipitation predictions, predictor fields, and attribution maps based on positive predicted precipitation events in western central Europe (index defined as the purple rectangle in a)). Panels show: (a, d) precipitation anomalies in Europe during OND for period P2 [1942-1969] and P3 [1970-2007], respectively; (b, e) attribution maps over the predictor field corresponding to positive events for periods P2 and P3, respectively. Panels (c, f) display global composites of OND anomalies in sea surface temperature and precipitation (shading), 200 hPa geopotential height (contours), and surface winds for positive events for periods P2 and P3, respectively. Attribution maps (b, e) indicate the relative contribution of each grid point in the predictor field to the forecasted value in the target region. The sum of the attribution values within each map equals the predicted anomaly in the corresponding index region (i.e., the sum of values in panel b) matches the precipitation anomaly within the purple box in panel a).

**Figure R6.** Composites of model anomalous precipitation predictions, predictor fields, and attribution maps based on negative predicted precipitation events in western central Europe (index defined as the purple rectangle in a)). Panels show: (a, d) precipitation anomalies in Europe during OND for period P2 [1942-1969] and P3 [1970-2007], respectively; (b, e) attribution maps over the predictor field corresponding to positive events for periods P2 and P3, respectively. Panels (c, f) display global composites of OND anomalies in sea surface temperature and precipitation (shading), 200 hPa geopotential height (contours), and surface winds for positive events for periods P2 and P3, respectively. Attribution maps (b, e) indicate the relative contribution of each grid point in the predictor field to the forecasted value in the target region. The sum of the attribution values within each map equals the predicted anomaly in the corresponding index region (i.e., the sum of values in panel b) matches the precipitation anomaly within the purple box in panel a).

**Figure R7.** (a) Climatology of SST (shading) and U200 (contours) for the common period of P2 (1942–1969) and P3 (1970–2007). (b) Differences between the climatologies of P2 and P3 for SST and U200, using the same representation as in panel a).

4. In the introduction it is stated that "The idea behind NN4CAST is to mitigate the risk of treating deep learning methods as "black boxes", thereby enabling users to identify sources of predictability and assess the sensitivity of predictions to variations in the training period and/or to the predictor region." (line 80). However, the current manuscript does not really analyze the sensitivity to the training period or predictor region.

We thank the reviewer for pointing out this issue. We agree that the current wording in the introduction may give the impression that the present manuscript directly analyzes sensitivities to the training period and predictor regions. To avoid this confusion, we will reformulate the text to clarify that these are functionalities that the NN4CAST framework enables in general, but that they are not explored in detail in the two case studies presented here. The focus of the current manuscript is instead on evaluating model skill and attribution patterns, while the broader flexibility of the framework will be emphasized more clearly as a potential for future applications. This point is now clarified in the revised manuscript (Lines 73-77).

**Specific comments:**

- 1. The description of how the ACC is calculated could be a little more clear on what dimension is being averaged over, spatially or temporally. For when it is spatial, it is also typical that an areal weighting is applied to account for latitudinal variations in grid area.
  - We thank the reviewer for this comment. In the revised manuscript, the figure captions now clarify how the ACC and RMSE are calculated, explicitly indicating whether correlations are computed across time (for spatial maps) or across space (for temporal series) (Figures 2 and 4). Regarding areal weighting, we have not applied it in the current plots. However, the model outputs are provided in a format that allows users to apply such weighting a posteriori if desired.
- 2. The different Listing's showing the python code are probably somewhat redundant. It would be more useful to show what architecture is implemented, which is not easily derived from the text. For example, there is an option for convolutional layers, but how is this implemented alongside the option for dense layers?

We thank the reviewer for the suggestion regarding the description of the network architecture, specifically the integration of convolutional layers with dense layers. In the revised manuscript, we have clarified this in the hyperparameter table (Tab. 1). Specifically, we now indicate that if at least one convolutional layer is applied, it is added at the input of the network, and its output features are flattened and concatenated with the dense layers, ensuring the combination of both convolutional and fully connected representations.

Additionally, all the Python listings previously included in the manuscript have been moved to the Supplementary Material. The main text now focuses on the description of the methodology and hyperparameters, while the Supplementary Material provides the detailed code examples for reproducibility.

3. Figure 4 and 6 colorbar is not uniformly spaced. Most of the values are near 0 or in the 0.2-0.4 range. Would be better to have separate colors for 0.2-0.3 and 0.3-0.4, to evaluate the skill.

We thank the reviewer for this suggestion. We have accordingly updated Figures 4 and 6, adjusting the color scales to better resolve the 0.2–0.4 range and improve clarity.

4. It is stated that linear statistical models are weakened by limited observational record and nonstationarity (line 369). However, it should be clear these are also limitations for the deep learning model.

We thank the reviewer for this comment. We have revised the sentence in the manuscript to clarify the advantages of NN4CAST. (Lines 373-378).

5. The subpanel titles in figures 3 and 6 are not clear. For example, "Regression predicted TNA on Niño years" could be something like "Input SST regressed against TNA index during El Niño"

We thank the reviewer for this suggestion. We have revised the subpanel titles in Figures 3 and 6 to make them clearer and more informative.

---

## Author Comment (AC2)

**Response to Reviewer 2**

October 20, 2025

**Detailed Comments**

Responses are marked in blue.

The manuscript presents NN4CAST, a Python framework intended to streamline seasonal predictability studies with deep learning. The pipeline covers data preprocessing (region/season selection, anomaly computation, trend removal), model construction with regularization, cross-validation and tuning, and interpretation via an XAI module and EOF analysis. Two case studies are used to illustrate skill: Pacific SST forcing of tropical North Atlantic (TNA) SST in boreal spring, and Pacific SST forcing of European autumn precipitation. The overall aim to facilitate testing sources of predictability and attributing predictions to input regions is very relevant to climate services, but the manuscript in its current form requires revision before it is suitable for publication.

**General comments**

1. The first case study (DJF tropical "Pacific" predictors-> MAM TNA SST) formally respects the lag, yet the predictor domain extends into the western tropical Atlantic. Given the well-known persistence of tropical Atlantic SST, even a narrow DJF Atlantic band can carry substantial memory into MAM and thus contribute to the high ACC shown in Fig. 2. In that sense, part of the reported skill may reflect local persistence rather than a Pacific-forced bridge. It would be helpful to clarify whether masking local Atlantic SST alters the ACC/RMSE/importance patterns.

We sincerely thank the reviewer for this highly relevant and constructive comment. We fully agree that, as you point out, even a narrow band of DJF SST in the western tropical Atlantic can carry substantial persistence into MAM, which in turn may artificially inflate the apparent skill in our first case study (DJF tropical Pacific predictors - MAM TNA SST).

As also raised by the first reviewer, we addressed this issue by designing an additional experiment in which we explicitly masked the predictor domain to exclude the Caribbean/western tropical Atlantic, while at the same time applying a complementary mask to the predictand field to exclude the Pacific. This setup ensures that there is no overlap between predictor and predictand regions, and thereby allows us to directly test to what extent local SST persistence may be influencing the results.

Importantly, this adjustment does not require any modification of the model code, since the masking can be implemented directly during the preprocessing of the SST fields prior to entering the prediction pipeline. The results of this sensitivity experiment show that model skill, measured both in terms of ACC and RMSE, remains high even after removing the Caribbean band from the predictor field (Figure R1). We do observe a modest reduction in skill in certain sub-regions (e.g., around the Gulf of Mexico), but the overall performance and attribution patterns remain consistent with those reported in the main text.

In the revised manuscript, we have updated the Figure 2 to illustrate these results, and we have also updated the Zenodo repository with the outputs of this new experiment to ensure full transparency and reproducibility.

Figure R1. Predictability of tropical North Atlantic SST variability from tropical Pacific anomalies. Panels showing model performance metrics over the full period (1901–2019) using a leave-one-out cross-validation approach for predicting the SST anomaly field of the tropical North Atlantic during MAM, with Tropical Pacific SST from DJF as the predictor. The predictions are compared against observed MAM SST anomalies. Specifically: (a) ACC spatial map, correlating at each grid point the observed and predicted time series (temporal dimension); (b) Time series of ACC maps, correlating for each year the observed and predicted spatial patterns (spatial dimension); (c) RMSE spatial map, computed analogously to (a); and (d) Time series of RMSE maps, computed analogously to (b), all calculated between predicted and observed fields. The ACC (RMSE) time series show the correlation (error) between predicted and observed global mean SST anomalies over time. Statistically significant results, determined using a one-tailed t-test at the 95\% significance level, are indicated by the non-dashed regions in panel (a) and values above the dashed line in panel (b).

2. In Fig. 3, the comparison between the regression composite for El Niño years (predicted TNA) and the importance composite should be improved. The fact that the attribution map contains cooling features does not, by itself, demonstrate added value, like mentioned in L280. It would help to explain how their sign and placement align with the atmospheric bridge and Wind-Evaporation-SST mechanism (e.g., stronger trades, surface heat-flux anomalies, wind-stress curl...), and whether the lead-lag structure supports that interpretation. As presented, it is difficult to separate a genuine teleconnection signal from collinearity in the SST field or residual Atlantic persistence. Showing that

attribution hotspots co-locate with observed flux/SLP/wind anomalies, and repeating the analysis with the Atlantic belt removed from the predictors, would clarify whether the cooling patterns reflect a physical mechanism or a model artifact.

We thank the reviewer for this valuable comment. We agree that the attribution patterns in Fig. 3 must be assessed in terms of their robustness and physical consistency. To address this concern, we repeated the experiment with the western tropical Atlantic masked from the predictor field, thereby removing the potential influence of local persistence, as explained in the previous comment. The results confirm that the central Pacific remains the dominant attribution hotspot, with only modest reductions in skill in certain sub-regions of the Atlantic basin, indicating that the cooling features identified in the attribution maps are not artifacts of Atlantic memory.

In addition, and following the reviewer suggestion, we investigated whether the attribution hotspots align with established physical mechanisms. We computed composites of anomalous surface winds, sea level pressure, and geopotential height at 200 hPa, conditioned on Atlantic indices (WTNA and SMSCU) as well as ENSO phases. In the central Pacific the model highlights regions of pronounced wind convergence and SST anomalies that lead to a Gill-type atmospheric response in upper levels (2 anomalous anticyclones at both sides of the equator), which projects eastward into the Atlantic as a Rossby-wave train. This teleconnection produces a negative NAO-like circulation pattern, characterized by a weakening of the North Atlantic subtropical high pressure system and associated trade winds, thereby affecting local surface fluxes and reinforcing SST anomalies in the tropical North Atlantic.

The co-location of attribution hotspots with these observed circulation features supports the interpretation that the attribution maps capture a physically meaningful teleconnection signal rather than statistical artifacts or collinearity in the SST field. To strengthen this point, we also computed analogous composites for negative phases of WTNA and SMSCU, which show consistent patterns but with opposite signs, further confirming the robustness of the mechanism.

In the revised manuscript we now present these new composites in Figure R2, while relocating part of the ENSO-conditioned analysis to the Supplementary Material. This separation highlights how the attribution patterns remain consistent across different conditioning approaches, while allowing readers to directly compare ENSO-based and Atlantic-index-based results.

**Figure R2.** Composites of model anomalous SST predictions, predictor fields, and attribution maps for positive predicted WTNA and SMSCU, based on 28 and 26 events, respectively. Panels a) and c) show the predicted mean SST anomalies in the Atlantic during MAM together with surface wind anomalies indicated by arrows. Panels b) and d) show the attribution maps over the predictor fields with SST in contours and surface winds in arrows. Panels e) and f) display global composites of MAM anomalies in sea level pressure (shading), 200 hPa geopotential height (contours), and surface winds for positive WTNA and SMSCU events. Attribution maps indicate the relative contribution of each grid point in the predictor field to the forecasted value in the target region, with the sum of the values within each map matching the predicted anomaly in the corresponding index region (i.e., the sum of values in panel c matches the WTNA anomaly within the purple box in panel a).

3. The manuscript often uses the language of "drivers," yet the analysis is primarily associational. This matters for teleconnections, where shared low-frequency covariates can produce strong correlations without isolating a pathway. The discussion around XAI (e.g., Fig. 3) therefore could be improved. For both applications, it would be helpful to control for NAO variability and check how strongly it modulates the two teleconnections, and how this impacts the

attribution maps. The recent literature arguing for causal-inference tools in teleconnection analysis points in this direction and could be useful (e.g. <a href="https://journals.ametsoc.org/view/journals/bams/102/12/BAMS-D-20-0117.1.xml">https://journals.ametsoc.org/view/journals/bams/102/12/BAMS-D-20-0117.1.xml</a>).

We thank the reviewer for raising this important point. We acknowledge that the term "driver" can be interpreted as implying causality, whereas our analyses are primarily associational. We will be more careful and use the term potential drivers when appropriate. In the first application (Pacific SSTs - Atlantic SSTs), our use of "driver" refers to the Pacific SST anomalies during DJF that precede and are statistically linked to Atlantic SST variability in the following MAM, indicating in this way a causality. Because the predictor and predictand are separated by a seasonal lag, the relevant information comes from the Pacific SSTs, while any NAO-related signal captured by the model is likely the component externally forced by SST rather than internally generated variability.

In the second application (Pacific SSTs - European precipitation), there is no lag between the predictor and the predictand field. Thus, the model primarily identifies statistical associations rather than direct physical causality. Here, surface atmospheric circulation is partially embedded in the anomalous behaviour of the SST field, which explains why attribution maps highlight significant SST contributions from extratropical regions in addition to the tropical Pacific. We understand that it is important to discuss the impact of the NAO when explaining precipitation in Europe. To analyze the impact of the NAO variability, we have explored composites conditioned on positive and negative NAO phases (Figure R3 for NAO+). They show how the NAO has a more internal variability in P2 (where no relationship appears with the SST anomalous field), while in P3 there is a clear association with ENSO.

**Figure R3.** Composites of observed anomalous precipitation and dynamical fields, based on positive observed NAO+ events in western central Europe (index defined as the purple rectangle in a)). Panels show: (a, c) observed precipitation anomalies in Europe during OND for period P2 [1942-1969] and P3 [1970-2007], respectively; (b, d) global composites of OND anomalies in sea surface temperature and precipitation (shading), 200 hPa geopotential height (contours), and surface winds for positive events for periods P2 and P3, respectively.

To explore the ENSO-NAO relationship further, Table R1 identifies the ENSO, NAO and precipitation cases conditioned, in periods P2 (1940-1969) and P3 (1970-2007) where we find some predictability in our model.

| Period    | Index  | Phase    | NAO
Negative | NAO Neutral | NAO
Positive | Total |
|-----------|--------|----------|-----------------|-------------|-----------------|-------|
| 1942–1969 | ENSO   | Negative | 3               | 4           | 3               | 10    |
|           |        | Neutral  | 4               | 4           | 4               | 12    |
|           |        | Positive | 3               | 1           | 2               | 6     |
|           |        | Total    | 10              | 9           | 9               | 28    |
|           | Precip | Negative | 4               | 3           | 3               | 10    |
|           |        | Neutral  | 3               | 4           | 4               | 11    |
|           |        | Positive | 3               | 2           | 2               | 7     |
|           |        | Total    | 10              | 9           | 9               | 28    |
| 1970–2007 | ENSO   | Negative | 5               | 5           | 5               | 15    |
|           |        | Neutral  | 5               | 3           | 2               | 10    |
|           |        | Positive | 3               | 4           | 6               | 13    |
|           |        | Total    | 13              | 12          | 13              | 38    |
|           | Precip | Negative | 5               | 6           | 1               | 12    |
|           |        | Neutral  | 5               | 5           | 4               | 14    |
|           |        | Positive | 3               | 1           | 8               | 12    |
|           |        | Total    | 13              | 12          | 13              | 38    |

**Table R1.** Contingency table of ENSO and precipitation index phases with NAO phases for the periods 1942-1969 and 1970-2007.

For P2, we can see that there is no clear relationship between ENSO and NAO, with half of the NAO events considered ENSO neutral (Table R1). However, for P3, 75% of NAO-like events were related to an ENSO phase of opposite sign (i.e., positive (negative) NAO-like is related to El Niño (La Niña)). This can be seen more clearly in Table R2 on conditional probability. In the P2 period, there is greater internal variability of the NAO, or in other words, more NAOs not associated with ENSO (p=0.42), therefore, in principle, not forced by SSTs in the tropical Pacific and less predictable in our model. However, in P3, the probability of finding NAO-like +/– associated with El Niño/La Niña increases significantly compared to neutral years (i.e. 0.45 in P3 vs 0.22 in P2, Table R2).

Regarding the impact of NAO+ and NAO- on precipitation, First of all, it should be noted that the precipitation pattern associated with the NAO (Figure R3) is very different from the first mode of precipitation variability in Europe identified by our model (Figure 5 in the manuscript). However, central Europe (i.e. the French box) appears to be affected in both maps.

The fact that NAO+/- in P3 are more strongly influenced by ENSO also has an impact on precipitation. The probability of positive precipitation associated with a NAO+ increases in P3 compared to P2 (i.e. 0.67 vs 0.29; Table R3) – Even so, 10% of precipitation variability is not explained by NAO (in P2 and P3, 0.29 and 0.30 respectively, probability of finding anomalous precipitation with neutral NAO). This stronger ENSO-NAO-like and

precipitation consistency in P3 is in line with the predicted precipitation anomalous maps (in Fig. 6 of the updated manuscript) and for the observed precipitation (Fig. R4). The composite of positive precipitation patterns in P3 shows a Niño-forced structure with negative SLP anomalies over Iberian Peninsula and British Isles (Fig. 6f and Fig. R4d). This pattern, although not strictly NAO positive, shares pressure anomalies in centres close to the canonical NAO index (Fig. R3d). However, the pattern in P2 shows a circumpolar wave that appears to originate in the Maritime continent region (with positive precipitation anomalies, which can induce an atmospheric wave, as explained in the text) and which projects onto the North Atlantic in a pattern distinct from the NAO (Fig. 6c and Fig. R4b compared with Fig. R3b). We have clarified all this analysis in the updated manuscript (Lines 361-366).

| Conditional probability               | 1942–1969 | 1970–2007 |
|---------------------------------------|-----------|-----------|
| p (Niño   NAO +)                      | 0.22      | 0.45      |
| p (Niña   NAO –)                      | 0.30      | 0.38      |
| p (ENSO neutral
(NAO + ∪ NAO –)) | 0.42      | 0.29      |
| p (NAO +)                             | 0.32      | 0.29      |
| p (NAO –)                             | 0.36      | 0.34      |

Table R2. Conditional probabilities between NAO and ENSO phases, based on the cases of Table R1.

| Conditional probability          | 1942–1969 | 1970–2007 |
|----------------------------------|-----------|-----------|
| p (NAO+   Pt+)                   | 0.29      | 0.67      |
| p (NAO-   Pt-)                   | 0.40      | 0.42      |
| p (NAO neutral
(Pt+ ∪ Pt-)) | 0.29      | 0.30      |
| p (Pt+)                          | 0.25      | 0.32      |
| p (Pt-)                          | 0.36      | 0.32      |

**Table R3.** Conditional probabilities between Precipitation index and NAO phases, based on the cases of Table R1.

**Figure R4.** Composites of observed anomalous precipitation and dynamical fields, based on positive observed anomalous precipitation events in western central Europe (index defined as the purple rectangle in a)). Panels show: (a, c) observed precipitation anomalies in Europe during OND for period P2 [1942-1969] and P3 [1970-2007], respectively; (b, d) global composites of OND anomalies in sea surface temperature and precipitation (shading), 200 hPa geopotential height (contours), and surface winds for positive events for periods P2 and P3, respectively.

4. The manuscript describes the toolkit as "versatile," yet for identifying dominant spatial modes it offers only EOF analysis of model outputs versus observations. For teleconnection work, this is a narrow diagnostic. At minimum, a versatile layer would include a menu of spatial-mode tools beyond EOF (e.g. maximal covariance or canonical correlation analysis for coupled patterns). Please consider either expanding the diagnostics accordingly or reframing the package as a DL-first pipeline with basic (EOF-based) spatial diagnostics.

We thank the reviewer for this valuable comment. We agree that for the analysis of teleconnections, EOFs provide only a narrow diagnostic. Our main objective with this toolkit, however, is to enable the modeling of teleconnections through deep learning. The EOF analysis is included primarily as a preliminary diagnostic, while a comprehensive implementation of additional techniques such as Maximum Covariance Analysis (MCA) or Canonical Correlation Analysis (CCA) falls outside the scope of the present work. We note that there are existing packages, such as xMCA (Rieger, 2021) ,Xcast (Hall et al., 2022) or Spy4Cast (Duran et al., 2024), that already offer these functionalities. We therefore see NN4CAST as complementary to such tools: outputs from the simulations

produced by our package can be readily used as inputs to these other libraries for further, more specialized, spatial-mode analyses. To avoid overstatement, we will revise the wording in the manuscript, with a clearer description that emphasizes this complementary role. We have clarified this in the updated version of the manuscript (Lines 6-8).

**Specific comments:**

L1 (...) with the changes in tropical sea surface temperatures (SST) being (...)

**We have corrected this.**

L8 Please be more specific than writing "(...) performs all the methodological steps".

**We have specified these steps. (Lines 8–10).**

L27 you already defined SST in the abstract. If you decide to define again, please use lower case as in the abstract.

**We have corrected this.**

L75-77 Here you introduce the tool for the first time, after a long introduction on seasonal forecasting and ML. I suggest bringing up the goal/what's new about your paper much earlier in the introduction, to help the reader to situate themselves.

We thank the reviewer for this suggestion. In the revised version of the paper, we have substantially reduced the introduction of Artificial Intelligence and Deep Learning, streamlining the background so that the reader can reach the goal and novelty of the paper much earlier.

L86 I don't think bringing up the possibility to combine NN4CAST with ESMValTool in the introduction is relevant. I think this could be mentioned in the conclusions/future work.

Thank you, we have moved this argument to the conclusions.

L89 Please give an example of such tools written in C/C++.

**Thank you, we have added as an example tool the Climate Data Operator (CDO)**

L92-103 I recommend not giving so many details of the applications to be analysed in this final introduction paragraph. Similarly, mentioning GitHub and code availability here seems misplaced.

We thank the reviewer for this comment. In the revised version, we have streamlined the final paragraph of the introduction, focusing on the main goal and contributions of NN4CAST without including detailed descriptions of the applications or code availability.

L337 I think the sentence should be rewritten, as significant skill is not found in most of the European continent, rather in parts of it.

We have modified the sentence according to the comment.

L375 and L380 are repeated

**We have corrected this.**

L380, L395 The authors mention that the tool has a primary application to identify windows of opportunity (WoO). However, in the two applications given, there was no framing related to WoO. I recommend improving the discussion towards the context of WoO.

We thank the reviewer for this observation. Following the suggestion, we have clarified the connection between NN4CAST outputs and the concept of windows of opportunity (WoO) in seasonal forecasting. In the European precipitation case study, period P2 (1942–1969) serves as an example of WoO, where the model shows high skill for predicting precipitation. By highlighting this period, NN4CAST can help identify time intervals where predictive skill is higher, providing useful insights for seasonal forecast applications. (Lines 402–406).

L388-L389 I suggest to focus on more specific advantages offered by the NN4CAST in your conclusions. "These complementary approaches offer valuable contributions to the scientific community and support the improvement of current seasonal forecasting systems" seems a bit vague and exaggerated at the same time. In particular for the first application, the authors did not go in depth to highlight any new insights concerning the teleconnection, rather used it as an example to illustrate what the tool does.

We thank the reviewer for this comment. Following the suggestion, we have revised the conclusion to emphasize the specific advantages of NN4CAST. The tool not only provides robust seasonal forecasts through cross-validation, but also systematically identifies the predictor regions contributing to target indices, quantifying their relative importance using explainable AI techniques. Importantly, the analysis of attribution maps allows linking predictive importance to known physical mechanisms, such as ENSO teleconnections. (Lines 402–408).

---

## Author Comment (AC3)

**Response to Reviewer 3**

October 20, 2025

**Detailed Comments**

Responses are marked in blue.

The paper introduces NN4CAST, a Python-based framework designed to identify and investigate drivers of seasonal climate predictability. It shows that NN4CAST provides explainability by attributing predictions to specific regions of the chosen predictor field, thereby quantifying the relative importance of different sources of predictability.

The paper addresses an interesting problem and proposes a framework for understanding sources of predictability. However, the manuscript currently lacks details on the method and justification of key choices, as well as on the interpretation of XAI results to make the framework truly useful for climate services and science. In the perspective of this reviewer, the framework as well as the examples chosen to illustrate its usefulness would benefit from some reconsideration prior to possible resubmission.

**General comments**

The method chosen to make the predictions is not discussed or justified in the paper. Why is an autoencoder architecture chosen in the first example? It should definitely be discussed whether this makes a difference to the regions identified by the XAI method? Given the short observational record and non-stationarity of the teleconnections, can a deep learning approach always be justified compared to a regularized regression?

We thank the reviewer for this insightful comment. In the revised manuscript, we have clarified the choice of network architecture and its advantages over simpler alternatives such as regularized regression. The structure we adopt (1024–256–64–256–1024) corresponds to a fully connected encoder–decoder, also referred to as an autoencoder-type MLP. This bottleneck design progressively compresses the high-dimensional predictor field into a compact latent representation before reconstructing the target field, which facilitates the extraction of the most relevant nonlinear predictive features while controlling model complexity.

To better situate our approach within the existing literature, we now explicitly reference studies that have applied autoencoder-type networks for seasonal prediction tasks. For example, Ibebychu et al. (2024) employed autoencoders combined with LSTMs to forecast ENSO, demonstrating the suitability of such architectures for capturing physically meaningful patterns in climate data. While our implementation is fully connected rather than recurrent, it follows the same encoder–decoder principle, ensuring interpretability and robustness when dealing with high-dimensional predictor fields. We have clarified this explanation in the text (Lines 238-246).

As for the short observational record and the non-stationarity of teleconnections, we acknowledge this as an important limitation. Our goal here is not to claim that deep learning will always outperform regularized regression, but to demonstrate that the NN4CAST framework is able to identify windows of opportunity and to capture skillful predictions even in challenging cases. For example, in the precipitation application, the framework reveals periods with significant skill despite the known non-stationarity of the ENSO-Europe teleconnection, something that would be difficult to capture with a purely linear model.

Ibebuchi, C. C., & Richman, M. B. (2024). Deep learning with autoencoders and LSTM for ENSO forecasting. Climate Dynamics, 62(6), 5683-5697.

This reviewer agrees with the two other reviewers that tropical Atlantic should not be included in predictor region in the first example.

As also raised by the first reviewer, we addressed this issue by designing an additional experiment in which we explicitly masked the predictor domain to exclude the Caribbean/western tropical Atlantic, while at the same time applying a complementary mask to the predictand field to exclude the Pacific. This setup ensures that there is no overlap between predictor and predictand regions, and thereby allows us to directly test to what extent local SST persistence may be influencing the results.

Importantly, this adjustment does not require any modification of the model code, since the masking can be implemented directly during the preprocessing of the SST fields prior to entering the prediction pipeline.

The results of this sensitivity experiment show that model skill, measured both in terms of ACC and RMSE, remains high even after removing the Caribbean band from the predictor field. We do observe a modest reduction in skill in certain sub-regions (e.g., around the Gulf of Mexico), but the overall performance and attribution patterns remain consistent with those reported in the main text.

In the revised manuscript, we have updated the corresponding figure (new Figure R1) to illustrate these results, and we have also updated the Zenodo repository with the outputs of this new experiment to ensure full transparency and reproducibility.

We sincerely thank the reviewer for this highly relevant and constructive comment. We fully agree that, as you point out, even a narrow band of DJF SST in the western tropical Atlantic can carry substantial persistence into MAM, which in turn may artificially inflate the apparent skill in our first case study (DJF tropical Pacific predictors - MAM TNA SST).

As also raised by the first reviewer, we addressed this issue by designing an additional experiment in which we explicitly masked the predictor domain to exclude the Caribbean/western tropical Atlantic, while at the same time applying a complementary mask to the predictand field to exclude the Pacific. This setup ensures that there is no overlap between predictor and predictand regions, and thereby allows us to directly test to what extent local SST persistence may be influencing the results.

Importantly, this adjustment does not require any modification of the model code, since the masking can be implemented directly during the preprocessing of the SST fields prior to entering the prediction pipeline.

The results of this sensitivity experiment show that model skill, measured both in terms of ACC and RMSE, remains high even after removing the Caribbean band from the predictor field. We do observe a modest reduction in skill in certain sub-regions (e.g., around the Gulf of Mexico), but the overall performance and attribution patterns remain consistent with those reported in the main text.

In the revised manuscript, we have updated the corresponding figure (new Figure R1) to illustrate these results, and we have also updated the Zenodo repository with the outputs of this new experiment to ensure full transparency and reproducibility.

Figure R1. Predictability of tropical North Atlantic SST variability from tropical Pacific anomalies. Panels showing model performance metrics over the full period (1901–2019) using a leave-one-out cross-validation approach for predicting the SST anomaly field of the tropical North Atlantic during MAM, with Tropical Pacific SST from DJF as the predictor. The predictions are compared against observed MAM SST anomalies. Specifically: (a) ACC spatial map, correlating at each grid point the observed and predicted time series (temporal dimension); (b) Time series of ACC maps, correlating for each year the observed and predicted spatial patterns (spatial dimension); (c) RMSE spatial map, computed analogously to (a); and (d) Time series of RMSE maps, computed analogously to (b), all calculated between predicted and observed fields. The ACC (RMSE) time series show the correlation (error) between predicted and observed global mean SST anomalies over time. Statistically significant results, determined using a one-tailed t-test at the 95\% significance level, are indicated by the non-dashed regions in panel (a) and values above the dashed line in panel (b).

Parts of the paper read a lot like a Python package documentation rather than a method or framework description (for example lines 156-164, Table 1, Listing 1-3). Since the paper is presenting a framework and not a package, this reviewer thinks that they might be better suited in the Appendix or Supplementary Material. In particular, the paper contains no details or discussion on the choice of deep learning method, which should be included in the main text - perhaps at the expense of the code description.

We thank the reviewer for this insightful comment. In the revised manuscript, we have moved all Python code listings (previously in the main text, including Listings 1–3) to the Supplementary Material. This change ensures that the main text focuses on the methodological framework and rationale rather than detailed code instructions. The full code and datasets used, are uploaded to the Github and Zenodo repositories, respectively.

Furthermore, we have expanded the main text discussion on the choice of deep learning methodology, including explanations of the network architecture (Lines 238-246). The main text now emphasizes the design decisions and reasoning behind NN4CAST, while detailed code examples for reproducibility are provided in the Supplementary Material.

In further agreement with the other reviewers, the results presented in Figure 3 c and d do not seem particularly convincing to this reviewer, and do not seem to highlight the value of model-based attributions. In the eyes of this reviewer, the composite importances identified by the XAI methods have very low amplitudes and don't show physically interpretable structure or coherence. How would the authors explain this? Furthermore, why is the data first filtered for El-Niño events, and how is the threshold chosen?

We thank the reviewer for this comment and acknowledge the concerns raised. Regarding the relatively low amplitudes of the attribution values, this is a consequence of the high spatial resolution of the predictor field: each grid cell represents one individual input of the model, so its contribution is necessarily small in magnitude. However, when aggregated across regions, these contributions add up to match the predicted signal. A potential extension, which we consider an interesting avenue for future work, would be to spatially aggregate attribution values into larger regions in order to better quantify their relative contribution to the overall forecast.

As for the filtering and thresholding criteria, the filtering was applied because our objective is to analyze interannual variability, which requires isolating this component from lower-frequency variability before the composites are computed. The threshold of  $\pm 0.5$  standard deviations was adopted consistently across all indices in order to include both moderate and strong events, thereby ensuring a larger and more representative sample size for the composites.

In the revised version of the manuscript, we now compute composites based on the predicted WTNA and SMSCU indices rather than conditioning exclusively on ENSO events. This change provides a more direct link between the model outputs and the attribution maps. In the revised version, we also include large-scale dynamical fields

(SLP, Z200, and surface winds) in the composites, which allows us to analyze how the attribution patterns relate to changes in atmospheric circulation in each case (see Section 4.2 of the new version).

**More specific comments**

Line 8: What do the authors mean by the 'original files'? Especially since this is in the abstract, a more specific term should be chosen.

We thank the reviewer for this comment. In the revised abstract, we have replaced the ambiguous phrase "original files" with "starting from the raw datasets" to clarify that NN4CAST operates directly on the unprocessed input data.

Line 59: It should be noted that this paragraph talks about Al models at weather timescales.

We thank the reviewer for this comment. We have clarified in the text that this paragraph specifically refers to AI models applied at weather timescales. (Lines 57–59).

Line 67: "The use of DL models to assess seasonal forecast is not so common" - Aside from the spelling error, this statement is very vague. Given the vast emerging literature on deep learning for seasonal forecasting, examples should be cited here, or the sentence should more specifically say what DL models have not been used for.

We thank the reviewer for this comment. In the revised text, we have clarified the statement and provided references to illustrate the limited use of DL in certain aspects of seasonal forecasting. Specifically, we now emphasize that most existing DL studies focus on individual phenomena or regions, rather than providing general-purpose, interpretable models that can handle multiple teleconnections. (Lines 60–67).

Line 132: It would be valuable to state why this method is chosen over others.

Thank you, we have clarified the use of this method due to its properties of sensitivity and implementation invariance, as stated by Sundarajan et al (2017). (Lines 123–125).

Sundararajan, M., Taly, A., & Yan, Q. (2017, July). Axiomatic attribution for deep networks. In International conference on machine learning (pp. 3319-3328). PMLR.

Line 133: "This method addresses the issue of non-linear problems, where the derivative of the output with respect to the inputs is not constant." This sentence is a bit too vague and slightly misleading - other XAI methods address non-linear problems as well, and Integrated Gradients can be applied to linear problems as well.

We thank the reviewer for this comment. In the revised text, we have clarified why Integrated Gradients (IG) is chosen, emphasizing its ability to provide axiomatic, theoretically grounded attributions that satisfy sensitivity and implementation invariance, which makes it particularly suitable for analyzing complex, high-dimensional climate predictors. We also revised the explanation to avoid implying that IG is uniquely applicable to non-linear problems, clarifying that it can be applied to both linear and non-linear models. (Lines 125–128).

Line 167: It is unclear to this reviewer what bullet point one intends to state. Furthermore, points 1-4 would be addressed by a regularized linear regression model as well - it would be valuable to include in this list why a deep learning approach is chosen here.

We thank the reviewer for this comment. We have added a clarification in the text emphasizing that, compared to a simple linear regression model, NN4CAST leverages deep learning to capture complex, nonlinear relationships and spatial interactions, which cannot be fully addressed by linear approaches. (Lines 158–170).

This reviewer is a non-English native speaker and appreciates the difficulties in writing in a second language. However, the paper would benefit from grammatical corrections, including but not limited to the following:

Line 1: 'being the changes in tropical sea surface temperature the most influential drivers'

Line 190 "By this way it avoids to introduce"

We thank the reviewer for this comment. We have carefully reviewed the text and corrected the identified grammatical issues, as well as other minor errors throughout the manuscript, to improve clarity and readability.